# Soft-matter-induced orderings in a solid-state van der Waals heterostructure

Kai Zhao[1,2,16], Baojuan Dong[1,2,3,16], Yuang Wang[4,16], Xiaoxue Fan[1,2], Qi Wang[4], Zhiren Xiong[1,2], Jing Zhang[1,2], Jinkun He[1,2], Kaining Yang[1,2], Minru Qi[2,5], Chengbing Qin[2,5], Tongyao Zhang[1,2], Maolin Chen[6], Hanwen Wang[7], Jianqi Huang[7], Kai Liu[8], Hanwei Huang[9,10], Kenji Watanabe[11], Takashi Taniguchi[12], Yaning Wang[13], Xixiang Zhang[6], Juehan Yang[14], Zhenwen Huang[15], Yongjun Li[15], Zhongming Wei[14]✉, Jing Zhang[1,2,3]✉, Shuoxing Jiang[4]✉, Zheng Vitto Han[1,2,3,7]✉ & Funan Liu[9,10]✉

Deoxyribose nucleic acid (DNA), a type of soft matter, is often considered a promising building block to fabricate and investigate hybrid heterostructures with exotic functionalities. However, at this stage, investigations on DNA-enabled nanoelectronics have been largely limited to zero-dimensional (0D) and/or one-dimensional (1D) structures. Exploring their potential in higher dimensions, particularly in combination with hard matter solids such as van der Waals (vdW) two-dimensional (2D) materials, has proven challenging. Here, we show that 2D tessellations of DNA origami thin films, with a lateral size over 10 μm, can function as a sufficiently stiff substrate (Young's modulus of ~4 GPa). We further demonstrate a two-dimensional soft-hard interface of matter (2D-SHIM), in which vdW layers are coupled to the 2D tessellations of DNA origami. In such 2D-SHIM, the DNA film can then serve as a superlattice due to its sub-100 nm sized pitch of the self-assemblies, which modulates the electronic states of the hybrid system. Our findings open up promising possibilities for manipulating the electronic properties in hard matter using soft matter as a super-structural tuning knob, which may find applications in next generation nanoelectronics.

Over the past decades, the highly controllable and atomically precise self-assembly of molecules has propelled the development of modern molecular-electronics. Among those known systems, the biological macromolecule DNA has a special yet sophisticated double helical chain-like structure at the nanoscale[1], which endorses a famed quasi-1D feature that inherently becomes a fascinating platform for constructing nanoelectronic devices[2–6]. Being categorized as a soft matter, DNA has been widely adopted in nanoscaled architectures to investigate the emerging mesoscopic physical properties. Notably, DNA self-assembly offers a highly programmable approach for constructing complex structures with high precision[7–9].

The nanometer-scale size and near atomic-level precision of DNA assemblies render them well-suited for the organization of nanoelectronics components for device manufacturing. For example, DNA assemblies have been extensively utilized as templates for nanoparticle patterning, as depicted in Fig. 1a, that facilitates the fabrication of various prototypic nanoelectronic devices including single electron transistor[10] and conductive nanowires[11,12]. Alternatively, the development of DNA-templated synthesis has enabled the site-specific patterning of various materials, including metals[13–18], metal oxides[13,18,19], metalloid oxides[15,20–23], biominerals[24,25], semimetals[26] and semiconductors[18], which paves the way to their

**Fig. 1 | Evolution of deoxyribose nucleic acid (DNA) assisted nanoelectronics.** **a** Schematic illustration of 0D-nanoparticle dressed DNA nanowires. **b** Schematic illustration of 1D-carbon nanotube supported by DNA nano-grooves. The arrays of carbon nanotubes are highly oriented thanks to the existence of nano-grooves. **c** The concept of next generation nanoelectronics based on a hybrid system of 2D-DNA origami and 2D van der Waals (vdW) materials. The 2D DNA origami tessellation can serve as **d** a moiré-like periodic superpotential, **e** a dielectric material in 2D field effect transistors (FETs), **f** or even an all-DNA FET, provided that each layer of DNA origami can be functionalized into metals, insulators, and semiconductors.

heterogeneous integration into complex nanostructures and nanoelectronic devices with precise spatial organization and tailored properties. More recently, self-assembled DNA nanotrenches were fabricated to spatially organize carbon nanotubes (CNTs) into highly-oriented arrays[27], as illustrated in Fig. 1b. This marked a breakthrough among a variety of applications in nanoelectronics based on DNA, as the CNT arrays can then be readily fabricated into a field effect transistor[28] – a cornerstone of modern logic circuitry, which has been dominated by solid-state hard matter semiconductors for decades.

So far, DNA nanoassemblies primarily serve as the template for the programmed organization of 0D and 1D nanoelectronic components. The exploration of their potential in modulating the electronic properties of 2D materials at a device level remains challenging. Additionally, their incorporation as a part of the functional entities in nanoelectronic devices has remained largely elusive.

In this work, we show that 2D tessellations of DNA origami thin films, with a lateral size of ~10 μm, can function as a free-standing substrate, whose Young's modulus is estimated to be ~4 GPa. The 2D DNA films are of a thickness of about 2 nm, faintly visible under optical microscope, and are of rather high air stability. These properties make them a promising candidate for constructing 2D nanoelectronics. We then demonstrate a two-dimensional soft-hard interface of matter (2D-SHIM), in which a 2D tessellation of DNA origami serves as a moiré-like superlattice and strongly modulates the electronic properties of the graphene layer placed atop. In particular, when the hybrid graphene/2D-DNA-origami devices are measured in the parameter space of magnetic field and carrier density $n$, additional sets of Landau fan are seen due to the sub-100 nm sized pitch of the self-assemblies of DNA origami. Our results demonstrate a unique platform by introducing an ordered 2D superlattice of DNA origami to vdW materials at a 2D interface, which opens up possibility for manipulating the electronic properties in hard matter using the spatial orders in soft matter as a super-structural tuning knob, which may shed lights on the potential applications in next generation nanoelectronics based on 2D-SHIMs.

## Results and discussion

### Synthesis and characterizations of 2D DNA films

The DNA origami technique[29] is a powerful nanofabrication method that enables the precise, programmable assembly of DNA molecules into complex nanostructures. It involves the folding of a bacteriophage-derived, long single-stranded DNA (scaffold) by hundreds of short synthetic oligonucleotides (staples) into designed shapes through Watson-Crick base-pairing. Rationally designed, geometrically compatible DNA origami nanostructures can be further joined together into tessellation patterns through base pairing and stacking interactions strategically positioned at their edges. With the continuous development of design principles and strategies, various tessellation patterns of DNA origami have been reported[30–40], with a few reaching a lateral size of a few μm. We here adopted a recently reported, optimized DNA origami tessellation system in which a critical design parameter (termed the interhelical distance) determining the conformation and tessellation capability of DNA origami units was identified and finely-tuned to minimize undesired curvature in monomeric DNA origami units, enabling the formation of diverse single-crystalline lattices ranging from tens to hundreds of square micrometers[34]. A detailed design workflow of the square DNA origami nanostructure for tessellation is provided in Supplementary Fig. 1. As shown in Fig. 2a, monomeric DNA origami units can be self-assembled from the DNA scaffold and staples, and further joined together into a square lattice with a periodicity of ~80 nm. A unit cell of such square lattice is presented in the solid black box, with the DNA joints highlighted in the purple circle inset in the top-right panel of Fig. 2a.

The as-prepared DNA origamis are dispersed in solution, and a delicate deposition process is needed to obtain flat and entire pieces of 2D DNA origami onto the target substrates. The DNA origami dispersed in the buffer solution was diluted with $MgCl_2$ solution (10.5 mM) and then dropped on the hard substrate. After holding for 1 h, the substrate with DNA solution on it was gently rinsed and blown dry with $N_2$ (Supplementary Fig. 2, see also Methods for more detail). A commonly used substrate to support DNA nanoflakes is mica.

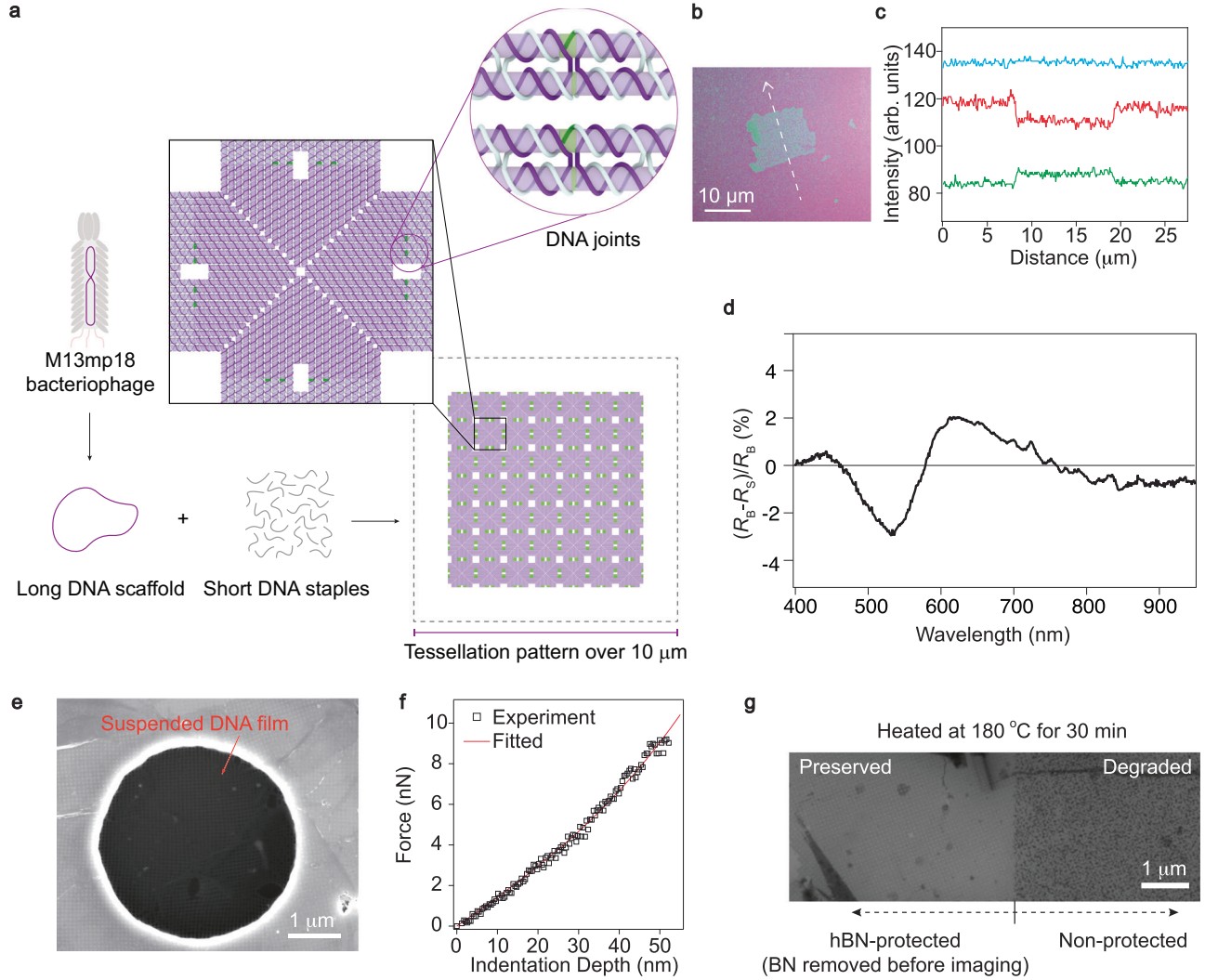

**Fig. 2 | Characterizations of tessellation pattern of DNA origami. a** Schematic illustration of the synthesis route of large size 2D DNA films by the tessellation of unit cells. **b** Optical image of a typical flake of micron-meter sized DNA film, deposited onto a SiO$_2$(300 nm)/Si$^{++}$ wafer. Scale bar, 10 μm. Notice that the optical contrast shown here is the blue channel extracted from an RGB format, which is named as LUT mode in the microscope (See Methods section). **c** Intensity profile along the white dashed arrow indicated in (**b**) plotted in Red, Green, and Blue channels recorded by the camera. **d** The renormalized reflectance difference (defined as $\frac{R_B - R_S}{R_B}$, where $R_B$ and $R_S$ are reflectance measured at the background and the sample areas, respectively) between the 2D DNA film placed onto SiO$_2$(300 nm)/Si$^{++}$ substrate and the substrate as a function of wavelength of the

perpendicularly incident light. **e** A drum-like 2D DNA film suspended on an etched hole on a Silicon (100) substrate. Diameter of the hole is 3.6 μm. **f** Typical curve of the tensile strain force versus indentation depth of a DNA drum shown in (**e**) The experimental data (black squares) were obtained by approaching an atomic force microscope (AFM) tip (see Methods section) onto the surface of suspended DNA film and then fitted by using a nonlinear force-displacement model[41] (red line). **g** Comparison between DNA film with and without h-BN protection, when subjected to a heating session in air. It is seen that after being heated at 180 °C for 30 min in air, the h-BN protected area is preserved, while the counter part without protection has been severely degraded.

Indeed, those DNA origami deposited on mica have highly ordered lattices, especially when the morphologies are characterized in liquid phase. Here in our work, 2D DNA films are aimed to be an ingredient substrate for 2D heterostructures. We therefore deposited DNA 2D films onto different substrates, including h-BN, graphene, SiO$_2$(300 nm)/Si$^{++}$ wafer, Al$_2$O$_3$, HfO$_2$, and mica (Supplementary Figs. 3 and 4). During the deposition process, the compatibility of DNA 2D films with various solvents has been tested as well (Supplementary Fig. 5). A typical micrograph image indicates that the tessellations of DNA 2D film have a weak visibility under optical microscope (Supplementary Fig. 6). Notice that Fig. 2b is obtained by modulating the original optical image's RGB channels (Supplementary Fig. 6), which significantly enhances the contrast of the DNA 2D film. And the intensity profile along the white dashed line in Fig. 2b is plotted in

Fig. 2c, where the reflection of red light is comparatively diminished, whereas the reflection of green light is relatively amplified. The normalized reflectance difference between the 2D DNA film placed onto a SiO$_2$(300 nm)/Si$^{++}$ substrate and the substrate itself as a function of wavelength of the perpendicularly incident light is presented in Fig. 2d (see Methods for more detail). This shows the same characteristic as in Fig. 2c that the red light reflected by DNA is weaker than the background and the green light is stronger than the background. The reflectance difference spectra of three typical DNA samples and monolayer graphene for comparison were additionally tested (Supplementary Fig. 7). The wavelength-dependent behavior of the DNA refractive index may offer insight into the mechanism by which the DNA superlattice interacts with light in the relevant spectral range (Supplementary Fig. 8).

We now examine the mechanical strength of such DNA 2D films. Holely substrates with diameters from 500 nm to 5 µm were first prepared via plasma etching (Supplementary Fig. 9). Depths of those holes are set to be above 1 µm. As shown in Fig. 2e, it is seen that DNA 2D films are stiff enough to be suspended onto the holes using a supercritical point dryer, while some of the films are partially suspended. Such drum-like suspended 2D films are capable to yield Young's modulus by being pressed using such as an atomic force microscope (AFM) tip, according to the simplified model[41]: $F = \pi\sigma_0 h\delta + \frac{Eh}{q^3 a^2}\delta^3$. Here, $F$ is the applied force, $\sigma_0$ is the tensile-stress in the suspended 2D film, $\delta$ is the indentation depth, $h$ is the effective thickness of the suspended DNA film (~2 nm), $E$ is the Young's modulus of the sample under test, $q = 1/(1.05 - 0.15\nu - 0.16\nu^2)$, where $\nu$ is Poisson's ratio (estimated to be 0.4 in our case), and $a = 3.6$ µm is the radius of the drum. Figure 2f shows a typical $F - \delta$ curve obtained experimentally in black and fitting line in red according to the simplified model. The Young's modulus of the DNA 2D film in this work is then fitted to be around 4 GPa (see also in Supplementary Fig. 9).

The air stability of the deposited DNA 2D films is also tested, when subjected to heating in air. It is seen that, shown in Fig. 2g, DNA

2D flakes can be degraded if heated in air at 180 °C for 30 min. SEM images of DNA origami heated in oxygen, nitrogen, and humid air (Supplementary Fig. 10) indicate that the degradation of DNA origami may be attributed to its structural disruption in the presence of water molecules when heated in air. However, such degradation can be avoided by covering the DNA film with an h-BN protecting layer. This allows us to further fabricate nano-electronic devices using the DNA origami superlattices (as will be discussed in the coming sections in more details), while keeping the integrity of them during the fabrication processes that often incorporates baking steps.

## Periodicity of the DNA 2D films

Figure 3a depicts the AFM height morphology of a typical square DNA 2D film deposited onto SiO$_2$(300 nm)/Si$^{++}$ substrate. It is seen that the tessellation of DNA origami forms a rather homogeneous 2D superlattice over a lateral size of a few µm. Some defect sites and/or dirts can be also seen. A zoomed-in scan in the boxed area in Fig. 3a is shown in Fig. 3b. The square lattice with a lattice constant of 84 nm is clearly shown, as plotted in the line profile in Fig. 3c along the dashed

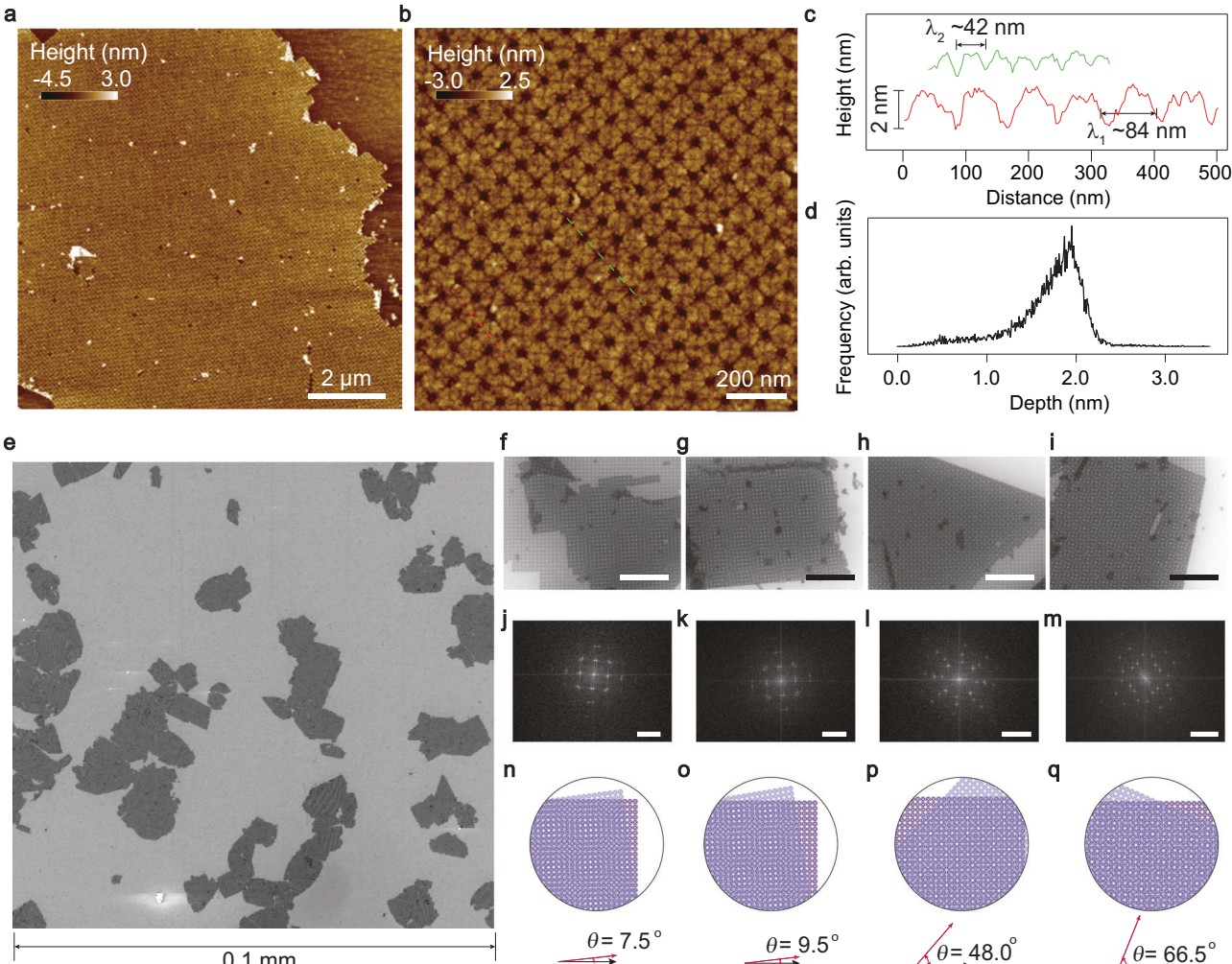

**Fig. 3 | 2D superlattice of DNA origami. a** Atomic force microscope (AFM) height morphology of a typical DNA 2D film deposited on SiO$_2$(300 nm)/Si$^{++}$ after being dried in air. **b** Zoom-in scan of a typical area in (**a**) **c** Line profiles along the red and green dashed lines in (**b**) where the periods of lattice ($\lambda_1$) and sub-lattice ($\lambda_2$) are marked. **d** The distribution of depth information for the surface area in (**b**) **e** Scanning electron microscope (SEM) image of the DNA 2D flakes deposited on SiO$_2$(300 nm)/Si$^{++}$, in a lateral range of 0.1 mm. **f**–**i** Moiré patterns formed by twisting two layers of monolayered DNA origami films at different twisting angles (7.5°, 9.5°, 48.0°, and 66.5°), with the corresponding fast Fourier transform (FFT) patterns shown in (**j**)–(**m**) and the cartoon illustrations shown in (**n**)–(**q**). The scale bars in (**f**)–(**i**) are 1 µm. The scale bars in (**j**)–(**m**) are 20 µm⁻¹.

red line in Fig. 3b. It is noticed that the tessellation also has a sub-lattice with 42 nm period, as plotted in the line profile in Fig. 3c along the dashed green line in Fig. 3b. Such DNA 2D flakes, after being dried in air, have a height of about 2 nm (Fig. 3d), which is slightly thinner than those measured under the liquid phase[34].

A direct illustration of the lateral sizes of the mono-dispersed DNA 2D flakes can be seen in Fig. 3e. The DNA films, if deposited properly, can be rather evenly distributed onto such as $SiO_2$(300 nm)/$Si^{++}$ wafers, and maximum size reaches over 10 μm. Interestingly, by chances some of the monolayered DNA 2D flakes can be self-folded with a certain twisting angle. This provides a unique opportunity to construct an additional set of moiré lattice. As shown in the SEM images in Fig. 3f–i, moiré patterns can be formed when two layers of DNA origami tessellations stack at a specific twisted angle. Fast Fourier transforms (FFT) were applied to analyze their spatial frequency distribution. Figure 3j–m are the FFT results of the SEM images in Fig. 3f–i. By analyzing the two sets of patterns, the twisted angles of the double-layer DNA tessellations can be inferred to be 7.5°, 9.5°, 48.0°, and 66.5°, respectively (see more details in Supplementary Fig. 11). The wavelengths of each moiré pattern varies from a few tens to a few hundreds of nm, as illustrated in Fig. 3n–q.

## DNA superlattice induced orderings in a 2D-SHIM

We now study the 2D-SHIM system by integrating DNA 2D film with van der Waals (vdW) materials, in order to investigate the super-potential created by DNA films. Taking graphene as an example, it is known that the band structure of the Dirac cone of graphene can be readily reconstructed due to the existence of an artificial super-lattice. The latter can be formed by the alignment of graphene/h-BN[42], etched hole arrays[43,44], and lithography-patterned lattices in the gate dielectrics[45,46]. A consequence of such superlattice (from 10 to few tens of nm in wavelengths) that is brought to graphene leads to the emergence of mini-fan at the finite doping away from the charge neutrality point (CNP) of graphene, in addition to the main Landau fan in the parameter space of magnetic field $B$ and carrier density $n$[42,47]. This is due to the fact that the moiré-potential superimposed to graphene lattice causes the formation of a mini-Brillouin zone in the $k$-space, which leads to an extra mini-band at finite energy from the charge neutrality (Supplementary Fig. 12). Usually, full filling of the mini-band corresponds to 4 electrons (taking the degeneracy of spin and valley for the cases of hexagonal moiré lattice) per moiré unit cell. That is $2.3 \times 10^{12}$ $cm^{-2}$ away from CNP for perfectly aligned graphene/h-BN[42].

In this study we mainly focus on the DNA 2D film with a square lattice shown in Fig. 3b. We now consider the design of a solid state heterostructure that can combine the soft-matter (DNA origami) and hard-matter (vdW layers) at the 2D interface, which is, a 2D-SHIM system. It is noticed that those DNA origami films are very difficult to be picked up like some vdW layers, impossible to adopt the dry-transfer stacking scheme[48]. One therefore has to seek the bottom up route to construct the heterostructure with the DNA origami non-transferred. Indeed, DNA film can be deposited onto the $Al_2O_3$ covered bottom gate in a pre-patterned manner, which is then located by SEM image (as shown in Fig. 4a) and is ready for vdW heterostructure deposition in later steps. The SEM-located DNA film is further aligned and deposited by an h-BN(20–30 nm)/graphene/h-BN(1-3L) stack, using the dry-transfer method[48] (Supplementary Fig. 13). Notice that the bottom gate dielectric $Al_2O_3$ is set to be 15 nm (Fig. 4c). The gate capacitive coupling is then through the ensemble of $Al_2O_3$(15 nm)/DNA(2 nm), which has strong enough gate modulation depth of electrostatic super-potential induced by the DNA superlattice, as confirmed by COMSOL simulations (Supplementary Fig. 14). Such a modulation of periodic potential is able to yield effective band reconstructions in graphene[46].

It is worth emphasizing that a 1-3L buffer h-BN is needed to avoid direct contact between graphene and DNA film, because of the possible scattering from charge traps that degrades the performance of the graphene transistor. Figure 4d shows the key findings of this work, i.e., the Landau fan of a typical graphene/DNA 2D-SHIM system (Sample-S18) plotted in $B$-$n$ parameter space at a temperature of $T$ = 1.5 K. It is seen that, except for the major fan (indicated by black lines extrapolating to $n$ = 0), two additional sets of fan-like features are observed, while being located at $\delta n \sim \pm 0.23 \times 10^{12}$ $cm^{-2}$ for each, as indicated by red lines in Fig. 4d. Figure 4g further plots the extracted representative branches of each Landau fan. To mitigate potential errors from manual identification, we assigned the Landau levels using equation $B = \frac{nh}{e\nu}$, where $n$ is the carrier density, $h$ is the Planck's constant, $e$ is the electron charge, and $\nu$ is the filling factor. Owing to the well quantized quantum Hall plateau in $R_{xy}$ (Supplementary Fig. 15), filling fractions can be quantitatively labeled for the left and major fans. However, the right fan, although clearly seen in $R_{xx}$, cannot be associated to its filling fractions due to the lack of quantization in the corresponding Hall resistance. We have emphasized that the $\delta n$ for the secondary Landau fan agrees well with the full filling of 4 electrons per unit cell (area $A \sim 1700$ $nm^2$), which may correspond to the square lattice with 42 nm period ($\lambda_2$) marked in Fig. 3c. A broadened main Dirac peak is also observed, which could be attributed to the merging of main Dirac peak with another set of Dirac peak in close proximity. This additional Dirac peak may arise from the 84 nm square superlattice ($\lambda_1$) marked in Fig. 3c, which should contribute to an additional $\delta n$ of approximately $\pm 0.05 \times 10^{12}$ $cm^{-2}$ (this density is however away too small to be resolved experimentally). To confirm the reproducibility of such square DNA origami induced additional Landau fan, we have collected data from another device (Sample-S15), which shows the similar electronic behavior (Supplementary Figs. 16 and 17). Furthermore, another geometry of DNA superlattice, composed of hexagonal DNA origami tiles with an edge length of 36 nm (Supplementary Fig. 18), was synthesized to demonstrate the modulation of the DNA superlattice on graphene's magneto-transport behavior. After optimizing the visibility of the DNA superlattice (Supplementary Fig. 19), a DNA/graphene device (Sample-S23) based on the hexagonal DNA origami 2D film has been fabricated and tested. As shown in Fig. 4e, two additional Landau fans indicated by green lines were located at $\delta n \sim \pm 0.12 \times 10^{12}$ $cm^{-2}$, agreeing well with the full filling of 4 electrons per unit cell area $A$ ($\sim 3400$ $nm^2$ for the hexagonal lattice with 36 nm edge length). These experimental results are in great agreement with the simulation results by solving the Hamiltonian derived from a continuum model (Supplementary Fig. 20 and Supplementary Note 2). While the estimated minimum $\delta n$ for the Landau fan induced by perfectly-aligned graphene/h-BN is about $2.3 \times 10^{12}$ $cm^{-2}$ away from CNP. The fan obtained from one of our control graphene/h-BN almost alignment sample is shown in Fig. 4f, which shows a typical $2.3 \times 10^{12}$ $cm^{-2}$ away from CNP for the Landau fan as highlighted by the blue lines. Here, it is noticed that the two side peaks appearing in a mirror-symmetric manner with respect to the main Dirac peak can rule out the inhomogeneity of doping in the sample. Additionally, linear $I$-$V$ curves (Supplementary Fig. 18d) as well as consistent multi-regional samplings (using multiple pairs of electrodes) of Landau fans (Supplementary Fig. 21) further confirmed negligible effects from unwanted doping variations. Brown-Zak oscillations (BZOs) are found to be also evidence for superlattice effects in some systems[49]. However, we regret that in our current study, no such BZOs were seen, which may be due to the relatively weak interaction between graphene and the DNA superlattice.

To conclude, based on the concept of soft-matter induced orderings in a 2D solid state heterostructure, we have devised a 2D-SHIM system, using DNA origami films with lateral sizes above 10 μm and 2D vdW materials as key ingredients. Solid state

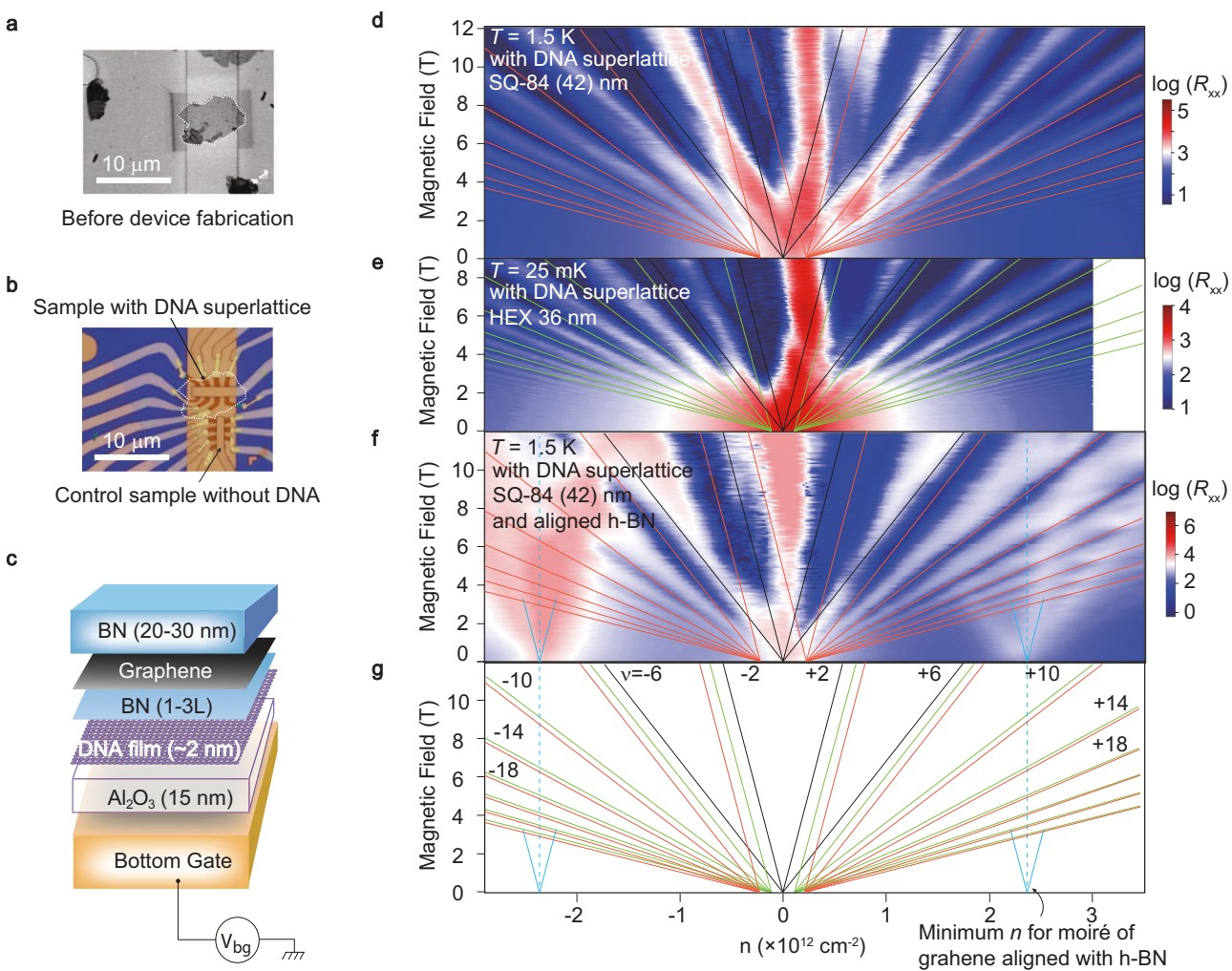

**Fig. 4 | A 2D interface of soft- and hard- matter. a** SEM image of the DNA origami 2D film on top of the pre-patterned bottom Au gate covered by 15 nm Al$_2$O$_3$. The DNA 2D film is outlined by the white dashed line. **b** Device fabrication of a typical vertical heterostructure of BN(20–30 nm)/graphene/BN(1–3 layer) on top of the DNA film in (**a**). The electrodes are patterned using standard lithography followed by metallization. Hall bars of the heterostructure are patterned via plasma etching. **c** Cartoon illustration of the vdW heterostructure. **d** Landau fan map ($\log(R_{xx})$) recorded in the parameter space of magnetic field $B$ and carrier density $n$) of a typical device with square DNA superlattice (Sample-S18) as shown in (**b**). Here $R_{xx}$ is the longitudinal resistance. **e** Landau fan map of a typical device with hexagonal

DNA superlattice (Sample-S23). **f** Control sample of a square DNA/h-BN/graphene/h-BN heterostructure with graphene and h-BN aligned at an angle close to zero degree. Samples in (**d**) and (**e**) are intentionally made to avoid alignment of graphene and h-BN during device fabrications. **g** The Landau Levels (LLs) (black: graphene; red: square DNA superlattice; green: hexagonal DNA superlattice; blue: aligned h-BN/graphene) extracted from (**d**) and (**e**) with the filling fractions labeled in each LL. The blue dashed line in (**f**) and (**g**) indicates the minimum carrier density for the origin of a Landau fan induced by almost perfectly aligned graphene/h-BN in our control sample, which is much higher than the observed mini-fans in (**d**) and (**e**).

heterostructure devices with 2D tessellation of DNA origami have been constructed. Systematic transport studies under magnetic fields up to 12 T and a base temperature of 1.5 K (or 25 mK) are further performed. It is evident that DNA film can serve as a superlattice due to its sub-100 nm sized pitch of the self-assemblies, which strongly modulates the electronic states of the hybrid system, by manifesting additional sets of Landau fans. Our findings suggest that DNA origami, a soft matter, can indeed be a powerful platform to effectively induce orderings in a solid-state heterostructure, which opens up possibilities to tailor the physical properties in future 2D-SHIM nanoelectronics.

## Methods
### Synthesis of DNA origami 2D films
The M13mp18 single-stranded DNA scaffold and the customized p3548 DNA scaffold strand were synthesized and purified following a

published protocol[34]. The DNA staple strands (sequences summarized in Supplementary Tables 1–5) were synthesized by GenScript (https://www.genscript.com.cn/) and received in the form of dry powder in 96-well plates. The staple strands were dissolved in DI water and quantified using NanoDrop One spectrophotometer (Thermo Scientific) by measuring the absorbance of each staple strand at 260 nm.

The staple strands were grouped based on their roles in the DNA origami nanostructure: core staple strands fold the scaffold strand into the designated geometry, while edge staple strands link monomer DNA origami nanostructures together into higher-order 2D films. The DNA origami 2D films composed of square tiles (wavelength = 84 nm) were prepared by mixing the M13mp18 scaffold (25 nM), core staple strands (250 nM/each), and edge staple strands (375 nM/each) in 1 × TAE/Mg$^{2+}$ buffer (40 mM Tris base, 20 mM acetic acid, 2 mM EDTA · Na$_2$, 12.5 mM magnesium acetate). The mixture was annealed from 80 °C to 20 °C in

~101 h using Mastercycler®nexus X2 PCR thermal cycler (Eppendorf). The annealing program was as follows: hold at 80 °C for 10 min, cool from 80 °C to 40 °C at −1 °C/min, hold at 40 °C for 10 min, cool from 40 °C to 20 °C at −0.1 °C/30 min, and hold at 15 °C until Step 2. In Step 2, reinforcing edge staple strands (375 nM/each) were mixed into the mixture. The mixture was heated up to 30 °C, held at 30 °C for 10 min, annealed from 30 °C to 20 °C at −0.1 °C/10 min, and held at 15 °C until use. The DNA origami 2D films composed of hexagonal tiles (edge length = 36 nm) were prepared by mixing the p3548 scaffold strand (25 nM), core staple strands (250 nM/each), and edge staple strands (375 nM/each) in $1 \times$ TAE/Mg$^{2+}$ buffer. The samples were annealed following the same procedure used for the square design.

### Deposition of DNA 2D films

The DNA origami 2D films composed of square tiles were dispersed in the $1 \times$ TAE/Mg$^{2+}$ buffer solution during transportation, and were diluted by mixing with MgCl$_2$ solution (10.5 mM) at a ratio of 1:39 before deposition. 15 µL of the diluted DNA solution were dropped onto O$_2$ plasma-cleaned Al$_2$O$_3$ (or other surfaces, Supplementary Figs. 3 and 4) and held for 1 hour. Then the deposited DNA films were gently rinsed with MgCl$_2$ solution, 70% ethanol, 80% ethanol and 100% ethanol solutions in sequence and blown dry with N$_2$. Different from the square superlattices, hexagonal superlattices were diluted by mixing with $1 \times$ TAE-Mg buffer solution (containing 12.5 mM Mg$^{2+}$) at a ratio of 1:3 before deposition, 3 µL of the diluted DNA solution were dropped onto the substrates and held for 30 min. Then the DNA films were gently rinsed and blown dry with the same methods as the previous square superlattice, except that MgCl$_2$ was replaced with $1 \times$ TAE-Mg buffer solution.

### Optical visibility

The LUT mode (Nikon LV-ND-100) was used to modulate the RGB channels of the optical image. The optical contrast was manually adjusted to maximum. SiO$_2$/Si$^{++}$ wafer with the oxide thickness of 300 nm was used as a substrate during the measurements.

### Differential reflectance spectroscopy of DNA 2D films

Differential reflectance measurements were performed using an LED white light source on a homemade microscope. The spectroscopy was demonstrated by measuring the difference in reflected intensity from the DNA/SiO$_2$/Si$^{++}$ wafer and bare SiO$_2$/Si$^{++}$ substrate and normalizing this to the SiO$_2$/Si$^{++}$ substrate-reflected intensity. The reflection lights were focused and collected by an objective (100× , NA = 0.9, NIRLT-APO, Attocube) and analyzed by a spectrometer (Andor, SR500I-A) with a cooled charge-coupled device (CCD, Andor, DR-316B-LDC-DD) to obtain the reflectance spectroscopy.

### Sample fabrication

vdW few-layers of the h-BN/graphene/h-BN sandwich were obtained by mechanically exfoliating high quality bulk crystals. The vertical assembly of vdW layered compounds were fabricated using the dry-transfer method in an ambient condition. The h-BN/graphene/h-BN sandwiches were then deposited onto the pre-fabricated DNA/Al$_2$O$_3$/Au structure, to form the complete structure as shown in Fig. 4c in the main text. Hall bars of the devices were achieved by plasma etching. During the fabrication processes, electron beam lithography was done using a Zeiss Sigma 300 SEM with an Raith Elphy Quantum graphic writer. Gate electrodes as well as contacting electrodes were fabricated with a electron beam evaporation, with typical thicknesses of Au/Ti -30/5 nm (or -8/2 nm).

### Young's modulus measurement

The tensile strain as a function of indentation depth curve is obtained using the PeakForce Tapping mode of the BrukerIcon AFM. RFESPA-40

or ScanAyst-Air tips are both used to measure the DNA drums, which yield similar results.

### Electrical measurements

Gate voltages on the as-prepared Hall bar devices were maintained by a Keithley 2400 source meter. During measurements, the graphene layer was fed with an AC $I_{bias}$ of about 100 nA (or 50 nA). Low-frequency lock-in four-probe measurements with a 13.3333 Hz excitation frequency were used throughout the transport measurements under high magnetic field and at low temperatures in an Oxford TeslaTron cryostat or a BlueFors dilution refrigerator.

## Data availability

The data that support the findings of this study are available via Zenodo at https://doi.org/10.5281/zenodo.14916148.

## Code availability

The code that support the findings of this study are available upon request to the corresponding authors.

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

## Acknowledgements

This work is supported by the National Key R&D Program of China (Nos. 2022YFA1203903 (Z.V.H.) and 2022YFA1305400 (S.J.)) and the National Natural Science Foundation of China (NSFC) (Grant Nos. 12450003 (Z.V.H.), 92265203 (Z.V.H.), 12104462 (H.W.), 12204287 (T.Z.), 12374185 (B.D.), U21A6004 (B.D.), 62125404 (Z.W.), 12034011 (J.Z., jzhang74@sxu.edu.cn), U23A6004 (J.Z., jzhang74@sxu.edu.cn), 62204145 (K.Z.), 22302092 (S.J.) and 12304237 (Y.W.)). Z.V.H. acknowledges the support of the Fund for Shanxi "1331 Project" Key Subjects Construction. Z.V.H., B.D., and J.Z. acknowledge the support of the Innovation Program for Quantum Science and Technology (grant no. 2021ZD0302003). J.Z. (jzhang74@sxu.edu.cn) acknowledges the support from the XPLORER Prize. X.Z. and M.C. acknowledge the support from King Abdullah University of Science and Technology's Office of Sponsored Research (OSR) under award No. ORA-CRG11-2022-5031. K.W. and T.T. acknowledge support from the JSPS KAKENHI (Grant Numbers 21H05233 and 23H02052), the CREST (JPMJCR24A5), JST and World Premier International Research Center Initiative (WPI), MEXT, Japan. Z.W. acknowledges support from the Beijing Natural Science Foundation (Z220005). Y.W. acknowledges support from the China Postdoctoral Science Foundation (2023M743579).

## Author contributions

F.L., Z.V.H., S.J., Z.W., and J.Z.(jzhang74@sxu.edu.cn) conceived the experiment and supervised the overall project. K.Z., B.D., H.H. and F.L. led the experiments of DNA film depositions, device fabrications, and electrical measurements; X.F., J.Z.(zj1752328218@gmail.com), H.W., and Y.W. helped in sample fabrications; Z.X., K.Y., J.Y., and J.H. (Jinkun) participated in electrical transport measurements; M.Q., C.Q. and T.Z. participated in the optical reflectance measurements; Q.W., Y.W., and S.J. synthesized the DNA 2D films; K.L., M.C., and X.Z. helped in suspending the DNA films onto holey Si substrate using a supercritical point dryer; K.W. and T.T. provided high quality h-BN bulk crystals; Z.H., and Y.L. contributed to the measurements of DNA film's elastic properties; Z.V.H., K.Z., B.D., S.J., J.Z.(jzhang74@sxu.edu.cn), F.L., and Z.W. analyzed the experimental data. J.H. (Jianqi) contributed to the band structure calculations. The manuscript was written by Z.V.H., K.Z. and B.D. with discussion and inputs from all authors.

## Competing interests

The authors declare no competing interests.

## Additional information

[1]State Key Laboratory of Quantum Optics Technologies and Devices, Institute of Optoelectronics, Shanxi University, Taiyuan, PR China. [2]Collaborative Innovation Center of Extreme Optics, Shanxi University, Taiyuan, PR China. [3]Hefei National Laboratory, Hefei, PR China. [4]State Key Laboratory of Coordination Chemistry, Department of Biomedical Engineering, College of Engineering and Applied Sciences, Nanjing University, Nanjing, Jiangsu, PR China. [5]State Key Laboratory of Quantum Optics Technologies and Devices, Institute of Laser Spectroscopy, Shanxi University, Taiyuan, PR China. [6]Physical Science and Engineering Division, King Abdullah University of Science and Technology, Thuwal, Saudi Arabia. [7]Liaoning Academy of Materials, Shenyang, PR China. [8]State Key Laboratory of New Ceramics and Fine Processing, School of Materials Science and Engineering, Tsinghua University, Beijing, PR China. [9]Department of Surgical Oncology and General Surgery, Key Laboratory of Precision Diagnosis and Treatment of Gastrointestinal Tumors, The First Hospital of China Medical University, Shenyang, PR China. [10]Phase I Clinical Trails Center, The First Hospital of China Medical University, Shenyang, PR China. [11]Research Center for Electronic and Optical Materials, National Institute for Materials Science, Tsukuba, Japan. [12]Research Center for Materials Nanoarchitectonics, National Institute for Materials Science, Tsukuba, Japan. [13]Shenyang National Laboratory for Materials Science, Institute of Metal Research, Chinese Academy of Sciences, Shenyang, PR China. [14]State Key Laboratory of Superlattices and Microstructures, Institute of Semiconductors, Chinese Academy of Sciences, Beijing, PR China. [15]Bruker (Beijing) Scientific Technology Co. Ltd, Beijing, PR China. [16]These authors contributed equally: Kai Zhao, Baojuan Dong, Yuang Wang. ✉e-mail: zmwei@semi.ac.cn; jzhang74@sxu.edu.cn; jsx@nju.edu.cn; vitto.han@gmail.com; fnliu@cmu.edu.cn

