## [Transparent Peer Review file · Nature Communications]

Soft-matter-induced orderings in a solid-state van der Waals heterostructure

Corresponding Author: Professor Zheng Han

Version 0:

Reviewer comments:

Reviewer #1

(Remarks to the Author)

The work by Zhao et. al. report stacking micron-scale DNA structures on two-dimensional materials, focusing on modulating the electronic properties of graphene by DNA structures. However, the core idea of the article is not novel in DNA field. DNA assembly and interacting with graphene lack innovation, and have been well established previously. And the mechanism that why DNA structures can tune the electronic structures of graphene is not sufficiently explained, lacking many important data to support the core idea of this work. The reviewer believes the technical basis of this work is not valid if these issues are not well clarified. The following are the main reasons why this paper should be rejected.

1. The authors seem not having a comprehensive understanding of DNA-enabled nanoelectronics. Using DNA structures to interact with graphene and other 2D materials is not new. More than 10 years ago, DNA structures have been reported to fabricate various shaped graphene structures and graphene electronics (for instance, Jin et al. Nature Comm. 2013, 4, 1663), which is not cited in this work. Compared to previous reports, the work here does not advance enough regarding integrating DNA structures with 2D materials. In addition, the authors draw a scheme in figure 2b showing that DNA structures are embedded between source-drain electrodes, which is different from their reference (In Ref. 26, DNA structures are eliminated before fabricating source/drain electrodes). These obvious but important mistakes should not appear in a paper submitted to Nature Communications.
2. There exist several important technical inaccuracies in the work. First, DNA structure contain high concentration metal ions, which are moving under gate bias. Thus, a very complex potential distribution will form considering this gate-driven ion movement. This potential distribution largely modulates the electronic states of graphene. It thus raises an import issue that what is the origin of DNA-modulated electronic states, phosphate backbones or metal ions? However, such effect is not discussed in the work. Second, previous research has been published using the nucleic acid structures to enhance the performance of two-dimensional material transistors. For instance, reference (D-H, Kang et al., Sci. Rep., 2016, 6, 20333) shows the impact of ion concentration and composition in nucleic acid structures on 2D materials (MoS₂, WSe₂), influencing the on-state current and threshold voltage of the transistors. However, the work here merely regards the DNA structure as a dielectric layer, overlooking the influence of metal ions on the potential distribution. Third, the authors report a super-moire pattern in Fig 3f-q. These patterns are often generated from the uncontrollable twisting of DNA structures, which induce different ion mobility and potential distribution. In Fig 4a, we could see that the DNA structure exhibit different twisting. So how will these different twisting patterns affect the electronic states? The authors need to supplement detailed analysis regarding the effect of twisting to the electronic states of graphene. If the authors could not provide any supportive experimental evidences, Fig 3f-q should be removed because of their uncontrollable twisting nature. Fourth, the reviewer believes Fig 1d-f should be removed from the figure, because these designs are technical inaccurate and misleading to the readers. For instance, periodic potential of DNA in Fig 1d is not valid considering metal ion movement, and all-DNA FET in Fig 1f is not relevant to current work. Fifth, the mechanism of electronic state modulation is not solid. More control experiments are needed regarding different DNA design (various thickness, periodicities and ion concentrations), 2D materials (h-BN, MoS₂ and WS₂), and environmental parameters (bias and temperatures). Theoretical simulations are also necessary.
3. The title of the paper suggests that materials with different mechanical stress properties are interacting at the interface, implying the possibility of new fusion techniques or novel physical properties at the interface. However, upon closer examination of the work, the process of DNA self-assembly and stacking DNA structures with two-dimensional materials

have been widely reported. There is no new physical mechanism observed at the interface of DNA soft matter and graphene hard matter.

4. Several typos in the References, for example Ref 26. and 29. The authors should double check and correct these.

Reviewer #2

(Remarks to the Author)

The reviewed manuscript presents interesting and systematic investigation of 2D tessellations of DNA origami thin films and corresponding 2D soft-hard interface of matter. The DNA origami thin films are confirmed to be sufficiently stiff substrate based on the mechanical strength experiment. The periodicity of the DNA 2D films is well defined and confirmed by AFM height morphology. The idea to create superlattice for hBN encapsulated graphene through periodic DNA origami thin films is of great novelty and pave a new way in exploring 2D-SHM nanoelectronics. However, I have a few concerns to be addressed.

1. The gap induced by the DNA superlattice is too close to charge neutrality in Fig. 4d. Is it possible to resolve the superlattice gap at zero field from the superlattice satellite peak?

2. The argument of DNA superlattice modulation is not convincing. First of all, the positions of superlattice gaps at hole and electron doping sides are not symmetric with charge neutrality, why? Secondly, the Landau level gaps are very broad making their trajectory difficult to be accurately traced. Any slight change of the slopes can give quite different conclusion. Moreover, if the local doping at different regions of graphene flake are different, the Landau Fan diagram can also show different sets of Landau levels as shown in Fig. 4d.

3. Is it possible to generate DNA superlattice with smaller periodicity so that the signatures of DNA superlattice can be better resolved?

Reviewer #3

(Remarks to the Author)

Patterning graphene with a spatially periodic potential provides a powerful means to modify its electronic properties [Nat. Nanotechnol. 16, 525 (2021); Nature Nanotech 13, 566 (2018); Nat Electron 4, 116 (2021)]. In previous works, the induced spatially potentials are usually "hard" materials. In this article, Zhao et al presented a novel strategy to alter the electronic properties of graphene by "soft" material, called DNA, which serve as a moiré-like superlattice after its self-assemblies. I think this work is very interesting, which can inspire more studies on the biology nanoelectronics. While I enjoy the results of this manuscript, there are some questions that should be addressed before I can recommend its publication. I detail my concerns below.

DNA periodic potential.

1. Formation of DNA period. DNA origami units can be self-assembled from a long DNA scaffold and hundreds of short DNA staples, and further joint together through base pairing and stacking into a square lattice with a periodicity of ~ 80 nm. May I ask this kind of assembled process is quite random or can be controlled? That is to say, the periodicity of ~ 80 nm can be further increased or decreased?

2. The profile lines in Fig.3b are not so clear, I suggest making the lines or colours a little bit thicker.

3. Super moiré. Usually, the super moiré is formed by stacking two more patterns together, and at least have three layers of materials. So, from my understanding, there only two layers of DNA stacking together with a twist angle, so how can we call supermoiré in Fig. 3f-i?

4. How to determine the twist angle in Fig. 3n-q? It is not so clear, and I suggest that the author make more analysis on the fast Fourier transformation (FFT) pattern shown in Fig. 3j-m.

5. The reproducibility of such DNA origami induced additional Landau fan, we have collected data from another device sample 15 (S15), which shows the similar electronic behaviour. I am very impressive that the authors have studied so many devices, I think it should be more than 15 devices. I am confused the main data of Fig. 4 come from which device? It seems that there are only two devices, not 15 devices. So how about the information of other devices? The author need make this clear.

Origin of mini-Landau fan diagram.

6. Fig. 4 is a little confused. In order to show the effect of DNA on the electronic property of graphene, the optical device of graphene without DNA and with DNA is shown in Fig. 4b, but the Fan diagram is just showing the graphene device with DNA (Fig.4d). It is better to show the fan diagram of graphene with and without DNA in the Fig. 4, like Fig.3 in Ref. [35]. If the author wants to further to claim the effect of DNA, similar with the effect of aligned G/hBN. Then we can also put the third maps in Fig. 4. For example, Graphene without DNA, Graphene with DNA and Graphene aligned hBN.

7. To demonstrate the period moiré potential effect, there are two effects: One is the second Dirac point, and the second the BZ oscillations which can survive high temperature. These two effects have been observed in precious moiré potential paper [Nat Commun 14, 4142 (2023)]. I suggest the authors cite this paper, and discuss why in current graphene/DNA system, there is no signature of Brown-Zak features.

8. The band structure of graphene can be reconstructed due to the existence of an artificial superlattice, and the DNA will change the band structure of graphene. So how the band structure will be changed? Such a modulation of periodic potential is able to yield effective band reconstructions in graphene. I suggest the authors need discuss a little about it at the end.

Version 1:

Reviewer comments:

Reviewer #1

(Remarks to the Author)

The revisions by Zhao et al. corrected few grammar and reference mistakes in the initial submission. However, unlike their statements in the rebuttal letters, the authors did not address the key technical concerns raised by the reviewer, i.e. the physical mechanism in Landau fan map. The proposed “DNA superlattice effect” still lacks enough experimental support, and was not consistent with the newly supplemented results, which are in line with the widely reported temperature effect, ion effect, and surface impurity effect. More importantly, the reviewer noticed that several figures of the work were likely re-used, labelled with different conditions, which raised the concerns over the data integrity. Based on these concerns, the reviewer strongly recommends to reject this manuscript.

Detailed technical and integrity concerns:

1. In Supplementary Figure 4, it is clear that Figure 4a (claimed to be after soaking in ethanol), Figure 4b (claimed to be after soaking in isopropanol solution), and Figure 4d (claimed to be after soaking in trichloromethane solution) have identical imaging details, noise level and sample defects, which are unlikely reproducible in different measurements. Similar issues could also be found in Supplementary Figure 4c and Figure 4e, where different processing conditions have been used. It seems that Supplementary Figure 4a, Figure 4d and Figure 4e are only parts of Figure 4b and 4c, with modified contrasts, height axis, and image dimensions, but labelled with totally different conditions. Another figure similarity occurs between Figure 4d in the main text (also Supplementary Figure 14a) and Supplementary Figure 17 c, where the authors claimed measured from different pin regions in Sample-S18. Despite different color settings and x-axis ranges were used in these figures, all the other figure details, including noise level and distribution, are identical. If all these figures are from one device, why the authors report the same figure three times?
2. The core conclusion of this work, as argued by the authors, is the “superlattice effect” of DNA structures on 2D materials. To support this conclusion, two experimental results are necessary, including the reproducible device construction to exclude random effect in sample preparation and the same-condition measurements of DNA samples with different periodicities/heights/surface morphologies to verify the effect arising from DNA molecules (even without superlattice) or the periodicities of DNA superlattice. The authors did provide a new device in Figure 4e. However, this new device (measured at 25mK) is not measured at identical condition to the other device (1.5 K), and cannot be used for direct comparison to support their conclusion on superlattice effect. Temperature effect plays a more important role here. The temperature effect could not be excluded here, and it combined with other effects discussed below. Therefore, it still lacks experimental evidence to confirm the superlattice effect.
3. The superlattice morphologies of DNA on real device samples are missing. Regarding the AFM images in Fig R8 and Supplementary Figure 15, these small-scale images (around 200 nm) do not reflect the lattice uniformity for the whole device (longer than 5 μm). Considering their DNA lattices adopt different morphologies and periodicities on different substrates, the true morphology on device is important. However, the authors did not indicate where the sample was measured. Considering DNA structures may be distorted with uncontrollable lattice morphology, without the high-resolution true morphology on a real device surface, it would be difficult to conclude if there was DNA superlattice effect.
4. More importantly, as shown in Figure 4a, the sample surface was covered with many assembly impurities (shown as the dark regions in the device areas in Figure 4b), which apparently affected the Landau fan diagram (as shown in Supplementary Figure 17). The same surface impurity would also appear, and deviate the Landau fan even when different DNA periodicities are used. The authors did not analyze these impurity effect, and ascribe the effects solely to DNA superlattice, which is not correct.
5. The authors argued that ion traps would lead to large hysteresis. However, this is not true. Considering the thin thickness of DNA, short-range movements (less than 1 nm) of ions did not introduce large hysteresis, this is different from those in bulk dielectric. In addition, discussion on DNA origami in electric field has been reported before (Li et al., ACS Nano 2015). Therefore, the simulation in Supplementary Figure 11 is technically wrong, because its model did not reflect the ionic nature of DNA. Besides, even though h-BN is used to block ions, these moving ions could still affect graphene transport, by generating strong electric field near graphene/h-BN surface, and did not need to be trapped in graphene lattice. Thus, the authors did not have enough evidence to exclude ion effect.
6. The authors believed that DNA superlattice effect was their key discovery. But such effect lacks supports, and even is contrary to the results. As shown in their experiments and discussed above, the authors did not consider the temperature effect, the impurity effect and ion effect, and did not provide enough evidence showing the large-area DNA morphology on real device surface. In addition, the key hypothesis of DNA superlattice effect was DNA structures as dielectrics (as shown in Figure 1e), which are directly contrary to previous reports using DNA as conductors (Fink et al., Nature 1999) and superconductors (Yu et al., Science 2001) at such low temperatures. The physical basis of this work is not solid. More importantly, the authors were using a geometry periodicity (30-40 nm), rather than the real sequence periodicity (100-200 nm) to fit their data. This is direct contrary to DNA superlattice effect. Because each nucleotide has distinct charge binding capability and surface electric field diffusion, their potential distribution could not be the same if their underlying sequences are different. As results, when DNA superlattice effect applies, the periodic potential will be sequence-determined, rather than morphology-determined (as used by the authors). So the periodicities used by the authors are just denying the proposed superlattice effect, and supporting a ion effect (which is consistent with geometric periodicity) mixed with impurity effect (consistent with Supplementary Figure 17).

Reviewer #2

(Remarks to the Author)

The authors have addressed all of my questions/comments. I recommend its publication.

Reviewer #3

(Remarks to the Author)

I have carefully reviewed the rebuttal letter, and the authors have thoroughly addressed my concerns.

The additional data on DNA origami deposition and heterostructure devices with a new-pitch DNA superlattice significantly strengthens the work. The authors have overcome major technical challenges, like integrating DNA deposition, van der Waals material transfer, stacking on DNA origami, and nanoscale device fabrication into a unified process—a big achievement. To my best knowledge, the combination of 2D materials with large-scale DNA origami and the demonstrated superlattice effect (graphene band structure engineering via DNA-induced super-potential) marks a groundbreaking advancement in DNA-based nanoelectronics with great potential.

I fully support the publication of this work in Nature Communications.

Version 2:

Reviewer comments:

Reviewer #4

(Remarks to the Author)

The study by Zhao et al. on the superlattice effect using DNA templates has significant academic value in two major aspects: 1) It enables the fabrication of high-quality nanostructures over a large area through a remarkably simple process, and 2) by designing the nanostructures as desired, it allows modulation of the quantum properties of 2D electronic devices.

The authors have put in extensive experimental efforts and actively addressed reviewers' concerns, ultimately improved the manuscript to a level suitable for Nature Communications. I would like to contribute to enhancing the quality of this paper by providing a few minor comments shown below:

1. Figure 2a presents a schematic of the tessellation pattern formed using DNA origami units. However, the explanation of the principle behind the formation of this tessellation pattern is highly insufficient. To understand the formation mechanism, readers are required to refer to ref. 33, which is inconvenient from the reader's perspective. Instead of describing the fabrication method, the paper should provide the fundamental principles or design rules governing the formation of the tessellation pattern.
2. Supplementary Figures 2–4 show the formation of DNA 2D films on various substrates. However, there is no discussion on why these experiments were conducted or what their results imply. First, the rationale behind substrate selection is missing—what differences were the authors aiming to observe? Additionally, while the results are presented, there is no discussion or conclusion regarding their implications. Were these experiments conducted to demonstrate thickness tunability? Or to show variations in the periodicity of the tessellation pattern? If the modulation of DNA films is influenced by the substrate, what is the underlying mechanism behind this phenomenon?
3. Figures 2b and 2c show OM images and color intensity of the 2D DNA film. However, the significance of these data is unclear. As presented, these figures merely indicate color variations in different regions, rather than demonstrating the presence of DNA. For example, in the case of graphene, the light-matter interaction results in an absorption of ~2–3% across most spectral ranges, allowing monolayer graphene to be identified via optical contrast in an optical microscope. Similarly, the paper should explain the mechanism by which the DNA superlattice interacts with light in the relevant spectral range, the extent of absorption, and how this leads to the observed optical contrast changes in OM images. Without this information, these figures appear unnecessary in the main text.
4. Figure 2g compares thermal treatment results with and without h-BN. However, it is unclear why the sample without h-BN undergoes degradation upon heating. Is it due to heat-induced deformation of proteins, or is it caused by structural degradation resulting from interactions with specific molecules in the air? A detailed explanation is essential.
5. The occurrence of mini Landau fans due to the superlattice effect is an important result. The authors demonstrate three different superlattice periods, showing that the carrier densities at which mini Landau fans emerge vary accordingly. However, adding the following two aspects would further strengthen the manuscript:
 - 1) The key interest here is how the energy band structure of graphene changes with different superlattice periods. While the authors discuss this based on the shift in the positions of the mini Landau fans, incorporating simulation results illustrating the corresponding modifications in the energy band structure would enhance clarity.
 - 2) Superlattice effects are generally known to become more pronounced under high back-gate voltages, as shown in the ref.

37, 38. To emphasize this aspect, a resistance vs. normalized carrier density plot for different back-gate voltages would significantly improve the argument (you can find in ref. 37, Fig 2 b). Given that the height of the fabricated superlattice (~2 nm) is relatively small, this measurement might be challenging. However, if mini Landau fans are observed, electrical measurements should still be able to capture the effect.

Version 3:

Reviewer comments:

Reviewer #4

(Remarks to the Author)

The authors have addressed all my comments. I believe that the current version of the manuscript is sufficient for publication in Nature Communications.

However, I noticed some minor typos in the newly added sections of the Supplementary Information. For example, in Supplementary Information Fig. S10, I found "Fig. R3b," and the sentence "the doping concentration is calculated to be $0.12 \times 10 \text{ cm}^{-3}$ as indicated in Supplementary Figure 19b" seems incorrect, as Supplementary Figure 19b is just an SEM image.

Before publication, please carefully review the entire Supplementary Information document.

Reviewer #1 (Remarks to the Author):

General Comment. *The work by Zhao et. al. report stacking micron-scale DNA structures on two-dimensional materials, focusing on modulating the electronic properties of graphene by DNA structures. However, the core idea of the article is not novel in DNA field. DNA assembly and interacting with graphene lack innovation, and have been well established previously. And the mechanism that why DNA structures can tune the electronic structures of graphene is not sufficiently explained, lacking many important data to support the core idea of this work. The reviewer believes the technical basis of this work is not valid if these issues are not well clarified. The following are the main reasons why this paper should be rejected.*

Response:

We appreciate very much the comments given by Referee#1.

However, we beg to differ her/his comments of “*the core idea of the article is not novel in DNA field. DNA assembly and interacting with graphene lack innovation, and have been well established previously*”.

As stated in the front page of this rebuttal letter, we found that the above assessment by our Referee#1 might be a misunderstanding due to the fact that we did not make a successful message-delivery of our core idea in the initial submission. We sincerely apologize for it, and list the novelty of our work below (new texts as well as references explaining our idea are now added in the Suppl. Info., highlighted in blue):

[REDACTED]

Fig. R1. Left: The parabolic energy dispersion for a free electron in k -space is given by the relationship between the energy E and the wave vector k . Right: Band structure formed due to the existence of a periodic potential (coming from the lattice of a natural crystalline of the atom sites, or from an artificial superlattice). Image is adapted from Ref. [R1].

^{R1} Edagawa, K. *Sci. Technol. Adv. Mater.* **15**, 034805 (2014).

1. The key idea of this work, is not simply doping graphene using some disordered DNA molecules with possible ions trapped in. Instead, what we are focusing is a ‘super-lattice’ effect, which can reshape the Brillouin zone and the Bloch waves vectors of a 2D electronic gas (for instance, graphene) placed on top of this specific superlattice.

The reason that a lattice (or a superlattice) is so important and different compared to only charge doping, is that a free electron with parabolic dispersion (*i.e.*, the relation of energy E and momentum k), when subjected to a periodic potential with proper wavelengths, can be described by a Bloch wave (illustrated in Fig. R1). This is the bedrock of the modern solid-state physics – band structure theories developed back in the 1950s.

2. As such, in a solid-state system, in order to demonstrate a superlattice effect, it requires a large enough sample due to two factors: on one hand, it requires a finite size to fabricate a mesoscopic device/transistor using lithography tools; on the other hand, in order to incorporate the superlattice effect, one has to have a periodic boundary condition that should repeat a considerably large number of periods in the lattice.

Fig. R2. (Cited as Supplementary Figure 9 in the updated Suppl. Info.) Comparison of trivial charge-impurity doping and the superlattice effect on graphene. a) Trivial charge-impurity doping and the consequence of Fermi level shifting of graphene without band reconstruction. b) Band reconstruction of graphene due to the existence of an artificial superlattice. Two side peaks can be seen due to the formation of additional minima in the DOS (which yield a resistance maximum accordingly, in the transport behaviors). For example, it can be realized by etching the SiO₂ substrate into holed arrays with a pitch of ~ 30 nm.^[R2]

^{R2} Forsythe, C., Zhou, X., Watanabe, K. *et al. Nat. Nanotechnol.* **13**, 566–571 (2018).

3. Difference between a trivial charge doping and the superlattice effect.

On one hand, we noticed that our Referee#1 mentioned (in her/his Comment-2) possible doping from the ions (according to Referee#1, such ions can be movable, when driven by gate) in the DNA. However, this kind of ion-traps will solely lead to a shift of Fermi level of graphene (illustrated in Fig. R2a), and sometimes give rise to large hysteresis and/or degradation of electron mobilities due to these short-range scatterings. We will come back to his issue when discussing in our response to Comment-2 given by Referee#1.

On the other hand, a superlattice effect will yield effective modification of band structures, giving rise to additional minima of density of states (DOS) at different energy. As illustrated in Fig. R2 (similar results can be seen from Fig. 2 in Ref. [R2]).

4. So far, superlattices in 2D nanoelectronics are mainly realized/defined via several means:

4.1 Twisted angle devices that exhibit moiré superlattice due to lattice mismatch of two flakes of 2D materials. For example, h-BN aligned with graphene, or double layered graphene with a twist angle against each other. This kind of superlattice is often generated by rotational operations of 2D crystals, which gives rise to band folding and additional DOS minima at higher energy in the reshaped band structure.

[REDACTED]

Fig. R3. Several typical superlattices by etching techniques. a) Lithography-etched arrays of SiO₂ substrate into holed arrays with a pitch of ~ 30 nm, which thus yields a periodic modulation of the electrical fields induced by the gate. As a consequence, electronic band structure of graphene placed above the holed arrays will be modified.^[R2] b) An h-BN/Graphene/h-BN is etched through an ordered array. Similarly, electronic band structure is modified, which generate a new pattern in the magneto-transport^[R3]. c) Focused ion beam etched template arrays.^[R4]

4.2 Etched lattices/arrays with precision tools, including electron beam lithography followed by reaction ion etching (shown in Fig. R3a-b), and focused ion beam (FIB) etched template used for patterning the ordered arrays in 2D materials (shown in Fig. R3c). These kinds of superlattices,

^{R3} Jessen, B., Gammelgaard, L., Thomsen, M. *et al. Nat. Nanotechnol.* **14**, 340 (2019).

^{R4} Ruiz, D., Sheinflux, H., Hoffmann, R. *et al. Nat. Commun.* **13**, 6926 (2022).

being either 2D or quasi-1D, are often realized by artificial micro-structure engineering, and can also give rise to band folding and additional DOS minima at higher energy in the reshaped band structure. **Because of the usually very weak superlattice effects, such band reconstructions are often manifested using the Landay fans with the existence of finite magnetic fields.** The corresponding transport evidence of band-reshaping for these cases listed in Fig. R3 are given in Fig. R4.

[REDACTED]

Fig. R4. Magneto-transport behavior of the samples shown in Fig. R3. a) The lithography-etched SiO_2 arrays lead to satellite fans in magneto-transport measurement.^[R2] b) Nonlinear Landau fan and a splitting of the zero-energy level independent of the magnetic field in an etched h-BN/Graphene/h-BN sample.^[R3] c) A rich Hofstadter butterfly observed in a FIB-etched graphene superlattice sample.^[R4]

5. Why we study DNA superlattices coupled to graphene.

As stated all above, novel types of superlattices are very much desired in 2D nano-electronics, since it can serve as a role of band structure engineering. Specific superlattices can even give rise to topologically non-trivial electronic bands (for instance, when C_{2z} symmetry is broken, twisted double layer graphene can manifest Chern insulators at zero magnetic field, which is of fundamental interests for condensed matter physics). Therefore, searching for new materials to be used as a superlattice is of great importance in this regard.

We notice that, instead of hard matter, new breakthrough in the synthesis of DNA origami, a soft matter, has been realized recently, and it can reach a 2D lattice over $10 \mu\text{m}$ in its lateral size – something not possible in the past. Our collaborative team thus aims at tackling the challenges of a new type of heterostructure made by interfacing 2D-DNA tessellations and graphene, and further explore the possibility of the above mentioned superlattice effect in the electronic behavior in graphene. We chose graphene because it is a well-known material with mature recipes in terms of device fabrication.

With the above five points, we would like to emphasize that, **the proposed soft-hard matter interface with the related superlattice effect in the transport in graphene, using 2D tessellation of DNA origami over $10 \mu\text{m}$ and periodicity of a few tens of nm, is a novel system that has never been studied before.** All the existing literatures reporting the so-called interaction

of DNA and graphene are limited with short range interaction and thus trivial doping or only morphology studies in most of the cases.

Our work is therefore a **first demonstration of superlattice effect** using long-range ordered 2D DNA origami as a platform. Our results suggest that such a soft matter meta-material can also play important roles in solid-state nanoelectronics. We hope that our detailed arguments listed above can convince Referee#1, and more importantly, can be beneficial for expanding the research scope of the community of not only DNA origami, but also the community of 2D materials.

Comment 1. *The authors seem not having a comprehensive understanding of DNA-enabled nanoelectronics. Using DNA structures to interact with graphene and other 2D materials is not new. More than 10 years ago, DNA structures have been reported to fabricate various shaped graphene structures and graphene electronics (for instance, Jin et al. Nature Comm. 2013, 4, 1663), which is not cited in this work.*

Response:

Indeed, we should cite this work, as it is a very good example to support the novelty of our work. (In our revised manuscript, it was now cited as Ref. [26] marked in blue color.) This 2013 Nature Communications paper shows that small-sized DNA structure (at the order of 100 nm) can act a template (etch mask) to etch graphene. However, the DNA structures in that study were solely used as templates in the processing stage and were removed afterwards. Moreover, it was limited to only morphology study, in their work, without any electronic transport measurements, likely due to the super small size of DNA structure at that time (10 years ago).

In contrast, our work introduces large-scale 2D tessellations of DNA origami as a functional component in device fabrication. These structures can be synthesized, deposited onto SiO₂ or other different wafer surfaces, and further fabricated into a device level equipped with gates and electrodes. In our graphene/DNA heterostructures, the DNA acts as an insulating layer that provides a periodic potential, resulting in a superlattice structure. This superlattice significantly modulates the overall electronic and magnetic transport properties of the device, as demonstrated by our systematic magneto-transport measurements in Fig. 4 in the main text in our revised manuscript, which is totally different from the results shown by Jin *et al.* Nature Comm. 2013, 4, 1663, mentioned by the Referee.

In the revised manuscript, we have updated our references, and have cited this work as Ref. 26. We thank again Referee#1 for her/his valuable discussion.

Compared to previous reports, the work here does not advance enough regarding integrating DNA structures with 2D materials. In addition, the authors draw a scheme in Fig. 1b showing that DNA

structures are embedded between source-drain electrodes, which is different from their reference (In Ref. 26, DNA structures are eliminated before fabricating source/drain electrodes). These obvious but important mistakes should not appear in a paper submitted to Nature Communications.

Response:

We appreciate very much this constructive suggestion by Referee#1.

We fully agree that in Ref. 27 (In our revised manuscript, the previously cited Ref. 26 has been updated and is now listed as Ref. 27 to maintain consistency with the updated citation list.), the DNA template was indeed removed as part of the fabrication process of aligned carbon nanotube transistors in their work. We apologize for this mistake in our initial submission.

In this new submission, we have, according to the suggestion of Referee#1, corrected the cartoon drawings in Fig. 1b in the main text.

We would like to sincerely thank Referee#1 for her/his careful examination of our manuscripts.

Comment 2. *There exist several important technical inaccuracies in the work. First, DNA structure contain high concentration metal ions, which are moving under gate bias. Thus, a very complex potential distribution will form considering this gate-driven ion movement. This potential distribution largely modulates the electronic states of graphene. It thus raises an import issue that what is the origin of DNA-modulated electronic states, phosphate backbones or metal ions? However, such effect is not discussed in the work. Second, previous research has been published using the nucleic acid structures to enhance the performance of two-dimensional material transistors. For instance, reference (D-H, Kang et al., Sci. Rep., 2016, 6, 20333) shows the impact of ion concentration and composition in nucleic acid structures on 2D materials (MoS₂, WSe₂), influencing the on-state current and threshold voltage of the transistors. However, the work here merely regards the DNA structure as a dielectric layer, overlooking the influence of metal ions on the potential distribution.*

Response:

We thank very much this discussion by Referee#1.

While appreciating the ‘ion-movement’ picture suggested by Referee#1, we would like to emphasize again that our major result is a superlattice effect, rather than trivial doping illustrated in Fig. R2. Contrary to those reported in D-H, Kang et al., Sci. Rep., 6, 20333 (2016).

More importantly, to exclude such influence of direct contact of any ions trapped in DNA with graphene, we have purposely inserted a very thin h-BN layer between the DNA superlattice and

the graphene in our devices, to keep the graphene in the as clean limit as possible. The superlattice of a Coulomb potential is realized by the modulation of electrical fields due to slight difference in the dielectric constant in the ordered DNA thin film. Such modulation of electrical fields is supported by our COMSOL simulations as given in Supplementary Figure 11. Similar results can be found in Ref. [R2-R4] (also in Fig. R3a and Fig. R4a).

As detailed in our response to the general comment in Page 2 in this rebuttal letter, we hope we now can convince our Referee#1 that charge/ion movement is not concerned in this work, and the proposed soft-hard matter interface with the related superlattice effect in the transport in graphene, using 2D tessellation of DNA origami over 10 μm and periodicity of a few tens of nm, is a novel system that has never been studied before.

Third, the authors report a super-moire pattern in Fig 3f-q. These patterns are often generated from the uncontrollable twisting of DNA structures, which induce different ion mobility and potential distribution. In Fig 4a, we could see that the DNA structure exhibit different twisting. So how will these different twisting patterns affect the electronic states? The authors need to supplement detailed analysis regarding the effect of twisting to the electronic states of graphene. If the authors could not provide any supportive experimental evidences, Fig 3f-q should be removed because of their uncontrollable twisting nature.

Response:

We thank very much this discussion by Referee#1.

It is a good question ‘*So how will these different twisting patterns affect the electronic states*’.

Our answer is we don’t know at this stage. Because these self-folded/twisted DNA flakes are often very small in their lateral sizes, which is super difficult to control and to be fabricated into devices, so far.

Here, we put these SEM images in the main text, only to show that the DNA 2D superlattice can be further folded into more versatile moiré-like patterns. This is more an inspiring aspect, or a kind of future possibility. If the Referee#1 insists, we could of course remove it eventually. But at this current version, we hope to keep them.

Fourth, the reviewer believes Fig 1d-f should be removed from the Fig., because these designs are technical inaccurate and misleading to the readers. For instance, periodic potential of DNA in Fig 1d is not valid considering metal ion movement, and all-DNA FET in Fig 1f is not relevant to current work.

Response:

We thank very much this discussion by Referee#1.

Since metal ion movement is not concerned in our work, we would like to show that long range ordered DNA 2D thin films can serve as a novel building block for constructing future nanoelectronics. Therefore, Fig. 1d-f is a kind of extension of the core idea of our work. We beg to keep them in the main text. And we thank very much for Referee#1's understandings and supports.

*Fifth, the mechanism of electronic state modulation is not solid. More control experiments are needed regarding different DNA design (various thickness, periodicities and ion concentrations), 2D materials (*h*-BN, MoS₂ and WS₂), and environmental parameters (bias and temperatures). Theoretical simulations are also necessary.*

Response:

We deeply appreciate Referee#1 for her/his thorough and professional suggestions.

Indeed, more control experiments are needed regarding different DNA design. As stated in the front page of this rebuttal letter, we have focused on optimizing the large-scale fabrication of DNA origami with smaller lattice constants (new samples with a hexagonal lattice, wave length = 36 nm is added in the revised manuscript) over the past few months, with particular attention to resolving issues like deformation during the rinsing process.

Unlike the relatively robust DNA superlattice composed of square DNA origami tiles (wavelength = 84 nm) used in our previously submitted manuscript, the new DNA lattice made of hexagonal DNA origami tiles (wavelength = 36 nm) was more susceptible to deformation and can thus easily disassemble. Fundamentally, the difference can be attributed to the matching rules that guide the inter-connection of monomer tiles (Fig. R5). In the square lattice, monomer tiles are periodically rotated, alternating their orientation throughout the lattice. This arrangement helps to cancel out any inherent structural deformation (curvature or twist, for instance) in individual tiles, ensuring overall planarity. The hexagonal lattice is composed of monomer tiles taking exactly the same orientation. This arrangement is found to be more susceptible to any minor deformation present in the monomer tiles.

Fig. R5. Compare and contrast the two types of DNA origami 2D lattices, a square lattice a) and a hexagonal lattice b), used in this study. Schematic illustrations depict the monomer tile and the matching rules defining the connections between tiles. Matching rules are denoted by Arabic numerals and asterisks, where n and n^* refer to two complementary edges. The relative orientations of tiles within each lattice are marked by yellow arrows.

Given that bivalent cations (e.g., 12.5 mM Mg^{2+}) are typically supplemented in the buffer recipe to neutralize the charge repulsion between DNA molecules and facilitate DNA nanostructure formation, changes in the ion condition can affect the packing density of DNA helices in the monomer tile, resulting in structural deformation in monomer tiles that can accumulate and deform the lattice. Our fabrication procedure involves consecutive rinsing steps that immerse DNA origami lattices in solvents lacking sufficient cations, which can induce deformation in monomer tiles. Therefore, compared to the square lattice, the hexagonal lattice is more sensitive and susceptible to such deformation due to its uniform tile orientation, which complicates its use in device fabrication. **To address this issue, we developed a tailored protocol for hexagonal lattice that enhances its planarity and integrity during the fabrication process.** Specifically, we reduced the incubation time for lattice deposition on silica or other hard-matter substrates to minimize residual contamination caused by solvent evaporation. We used $1\times$ TAE-Mg buffer (containing 12.5 mM Mg^{2+}) for subsequent rinsing steps to prevent lattice deformation induced by changes in ion conditions.

Furthermore, by investigating the effect of substrate roughness used for DNA deposition, our study reveals that smoother surfaces can lead to better preservation of DNA periodicity. Specifically, we compared the periodicity of DNA deposited on Al_2O_3 film (obtained by atomic layer deposition), which are grown on smooth $\text{SiO}_2/\text{Si}^{++}$ substrates and a Au/Ti ($\sim 50/5$ nm) local gate pre-fabricated on a $\text{SiO}_2/\text{Si}^{++}$ substrate. The latter, Ti/Au deposited by electron-beam evaporation, is supposed to exhibit larger roughness. Indeed, SEM images shown in Fig. R6 indicate that less distinguishable DNA periodicity is seen on the Al_2O_3 film inherited from the Ti/Au local gate compared to that deposited directly on $\text{SiO}_2/\text{Si}^{++}$ surface. This demonstrates the critical role of substrate roughness in maintaining reasonable periodicity of DNA structures during the sample fabrication processes.

Fig. R6. DNA origami 2D film on $\text{Al}_2\text{O}_3/\text{SiO}_2/\text{Si}^{++}$ and $\text{Al}_2\text{O}_3/\text{Au}/\text{Ti}/\text{SiO}_2/\text{Si}^{++}$ substrates. a) and b) are the schematic diagram and scanning electron microscope (SEM) image of DNA origami 2D film on $\text{Al}_2\text{O}_3/\text{SiO}_2/\text{Si}^{++}$ substrate. c) The zoomed-in view of the circled area in b). d) and e) are the schematic diagram and SEM image of DNA origami 2D film on $\text{Al}_2\text{O}_3/\text{Au}/\text{Ti}/\text{SiO}_2/\text{Si}^{++}$ substrate. f) The zoomed-in view of the circled area in e).

To achieve better surface smoothness, we optimized the thickness of the local gate. Comparative SEM analysis of substrates with varying local gate thicknesses revealed that thinner local gates exhibited smaller surface grain roughness, leading to a better resolution of DNA periodicity. The corresponding results are presented in Fig. R7. As mentioned in the ‘Sample fabrication’ section of our previous submitted manuscript, the buried electrodes in two successful devices showcasing additional Landau fans (Sample-S18 and Sample-S15) were both composed of Au/Ti ($\sim 30/5$ nm). Fig. R7b shows that with this kind of buried electrode, the periodicity of DNA can still be observed.

Here, we have set the thickness of the buried electrodes to thinner Au/Ti ($\sim 8/2$ nm) electrodes to fabricate new device with hexagonal DNA origami (wavelength = 36 nm) 2D film.

Fig. R7. The influence of the metal layer thickness beneath Al_2O_3 on the visibility of DNA origami 2D film. SEM images of DNA origami 2D film deposited on Al_2O_3 films with buried metal gate electrodes of Au/Ti ($\sim 8/2$ nm) a), Au/Ti ($\sim 30/5$ nm) b), and Au/Ti ($\sim 50/5$ nm) c). The schematic diagrams at bottom of each SEM images illustrate the thickness of the buried metal electrodes.

New DNA/graphene devices based on hexagonal DNA origami 2D film (wavelength = 36 nm) have been fabricated and tested. Additional Landau fans besides the main fan in the device (Sample-S23) was observed (shown in Fig. R8d). As shown in Fig. R8a-b, the atomic force microscope images show that the DNA origami 2D film has a honeycomb distribution with a 36 nm periodicity. A standard Hall-bar device (Fig. R8c, Sample-S23) was prepared and corresponding magneto-transport was studied. Fig. R8d (also cited as Fig. 4e in the main text of our revised manuscript) demonstrates the superlattice effects induced by the DNA origami. We assigned the index of Landau levels using the equation $B = nh/ev$, where n is the carrier density, h is Planck's constant, e is the electron charge, and v is the filling factor. This approach minimizes subjective bias and enhances the accuracy and reliability of our Landau level labeling. As shown in Fig. R8d, three sets of Landau fans were assigned. Among these, two symmetric Landau fans (depicted by the green lines) are positioned at $\pm 0.145 \times 10^{12} \text{ cm}^{-2}$ away from the main Dirac fan (depicted by the black lines), agreeing well with the full filling of 4 electrons per unit cell (area A

$\sim 2700 \text{ nm}^2$), which agrees with the surface area of a unit cell of the studied hexagonal DNA lattice with 36 nm wavelength (Fig. R8a-b).

Fig. R8. The magneto-transport behavior of the graphene/DNA heterostructure with a hexagonal DNA superlattice (wavelength = 36 nm). a) AFM characterization of the 36 nm hexagonal DNA origami 2D film. b) Zoomed-in scan of the area outlined by the green dashed box in a). The scale bars in a) and b) are 100 nm. c) Optical image of the device (Sample-S23) and schematic of the measurement setup. d) Landau fan obtained from the graphene/DNA heterostructure (Sample-S23). The solid black lines indicate the Landau levels for the main Dirac fan of graphene, while the green lines represent the additional side Dirac fans originating from the hexagonal DNA superlattice. The satellite peaks are positioned at $\pm 0.145 \times 10^{12} \text{ cm}^{-2}$ away from the charge neutrality point (CNP) of graphene, as indicated by the extensions of the green lines.

Comment 3. *The title of the paper suggests that materials with different mechanical stress properties are interacting at the interface, implying the possibility of new fusion techniques or*

novel physical properties at the interface. However, upon closer examination of the work, the process of DNA self-assembly and stacking DNA structures with two-dimensional materials have been widely reported. There is no new physical mechanism observed at the interface of DNA soft matter and graphene hard matter.

Response:

We thank very much this point raised by Referee#1.

However, as discussed in the previous threads, we beg to differ with the comment ‘*DNA self-assembly and stacking DNA structures with two-dimensional materials have been widely reported. There is no new physical mechanism observed at the interface of DNA soft matter and graphene hard matter*’.

In fact, all the existing literatures reporting so-called interaction of DNA and graphene are limited with short range interaction and thus trivial doping or only morphology studies in most of the cases. From a condensed matter physics point of view, novel types of superlattices are very much desired in 2D nano-electronics, since it can serve as a role of band structure engineering. Specific superlattices can even give rise to topologically non-trivial electronic bands (for instance, when C_{2Z} symmetry is broken, twisted double layer graphene can manifest Chern insulators at zero magnetic field, which is of fundamental interests for condensed matter physics). Therefore, searching for new materials to be used as a superlattice is of great importance in this regard.

Our work is a **first demonstration of superlattice effect** using a heterostructure with long-range ordered 2D DNA origami and graphene as a platform. Our results suggest that such a soft matter meta-material can also play important roles in solid-state nanoelectronics, which opens new possibilities in electronic band engineering via an interdisciplinary route. We believe that a lot of challenges as well as opportunities are yet to come in this filed.

Meanwhile, our title indeed may be interpreted as an imply of ‘possible new fusion techniques’. In this regard, we fully agree with the Referee#1, and have modified the title into:

“Soft-matter induced orderings in a 2D solid state heterostructure”.

Comment 4. *Several typos in the References, for example Ref 26. and 29. The authors should double check and correct these.*

Response:

We appreciate the very careful readings of Referee#1.

We sincerely apologize for the typos in the references, and have corrected them in the new submission.

Finally, we would like to sincerely thank the very helpful comments given by Referee #1. The revisions according to her/his suggestions have made our manuscript of much improved quality. Her/his support in publication in Nature Communications will be greatly appreciated.

Reviewer #2 (Remarks to the Author):

General Comment. *The reviewed manuscript presents interesting and systematic investigation of 2D tessellations of DNA origami thin films and corresponding 2D soft-hard interface of matter. The DNA origami thin films are confirmed to be sufficiently stiff substrate based on the mechanical strength experiment. The periodicity of the DNA 2D films is well defined and confirmed by AFM height morphology. The idea to create superlattice for hBN encapsuled graphene through periodic DNA origami thin films is of great novelty and pave a new way in exploring 2D-SHIM nanoelectronics.*

Response:

We thank the reviewer for her/his positive and encouraging comments on our work. We are delighted that the novelty and systematic investigation of 2D tessellations of DNA origami thin films, as well as the resulted 2D soft-hard interface of matter (2D-SHIM), were well-appreciated by Referee#2. We are especially grateful for Referee#2's recognition of our innovative approach in leveraging periodic DNA origami 2D films to create a superlattice.

We sincerely appreciate the reviewer's valuable suggestions, which have significantly improved the quality of our work.

In the following, we address each of her/his comments in a point-to-point manner.

Comment 1. *However, I have a few concerns to be addressed.*

The gap induced by the DNA superlattice is too close to charge neutrality in Fig. 4d. Is it possible to resolve the superlattice gap at zero field from the superlattice satellite peak?

Response:

We appreciate the reviewer's insightful question regarding the resolution of the superlattice gap and its relationship with the superlattice satellite peaks at zero field. First, we must admit that the superlattice effects in this current work is indeed rather weak, so that we manifested such property with the existence of a finite magnetic field, as the side peaks are less developed in the absence of magnetic field, as pointed out by our Referee#2. To address this, we performed Lorentzian fitting of the longitudinal resistance peaks observed in the line-cut profiles shown in Fig. R9 at zero magnetic field. The fitted peak positions of the superlattice satellite peaks correspond to the carrier concentrations associated with the DNA superlattice periodicities.

For our devices with DNA superlattice periodicities of 36 nm and 42 nm, the carrier concentrations derived from the fitted peak positions align well with the theoretical values predicted by the relation $n=1/(eS)$, where S represents the unit cell area defined by the superlattice periodicity. This agreement between the Lorentzian-fitted peak positions and the expected carrier densities demonstrates that the DNA superlattice effectively modulates the graphene band structure.

Although resolving the superlattice gap at zero field is challenging due to the weak coupling strength and potential interfacial disorder, the Lorentzian fitting provides a suitable supplementary method for identifying the superlattice satellite peaks and confirms their correspondence to the superlattice periodicity. The line-cuts and fitted curves are presented in Fig. R9, showing a clear match between the experimental data and fitted profiles.

Fig. R9. The line-cut profiles and fitting of longitudinal resistance for graphene at zero magnetic field.

The red and yellow circles represent the resistance measured from graphene interfaced with 36 nm (Sample-S23) and 42 nm (Sample-S18) periodic DNA templates, respectively, while the blue circles correspond to pure graphene encapsulated by h-BN. The yellow (green) solid line represents the fitted curve for the main graphene Dirac peak, while the purple (pink) and blue (orange) lines correspond to the fitted curves for the superlattice satellite peaks. And the solid lines overlaying the circles are the combined fitting curves, obtained by summing all individual fitting components.

Comment 2. *The argument of DNA superlattice modulation is not convincing. First of all, the positions of superlattice gaps at hole and electron doping sides are not symmetric with charge neutrality, why?*

Response:

We appreciate the reviewer's careful reading regarding the slight asymmetry in the positions of the superlattice gaps relative to charge neutrality. Upon carefully re-evaluating our Landau levels, we found that the initial slight asymmetry was due to inaccuracies by manual reading of the lines of the Landau fan, which have now been corrected. To mitigate potential errors from manual identifications, we assigned the Landau levels using the equation $B = nh/ev$, where n is the carrier density, h is Planck's constant, e is the electron charge, and ν is the filling factor. This approach minimizes subjective bias and enhances the accuracy and reliability of our Landau level labelling. In the revised manuscript, we have updated Fig. 4d-f to accurately represent the symmetric positions of the superlattice gaps. Additionally, we have included the detailed method used for assigning the Landau levels in our revised manuscript in blue color. We would like to thank our Referee#2 for this very helpful suggestion.

Secondly, the Landau level gaps are very broad making their trajectory difficult to be accurately traced. Any slight change of the slopes can give quite different conclusion.

Response:

We appreciate the reviewer's insightful observation regarding the broadness of the Landau level gaps, particularly at low filling factors, which indeed poses a challenge for accurately tracing their trajectories. This is a valid concern, as the broad gaps at low filling factors make precise Landau level identification more difficult. Here, according to the equation $\Delta E = Be\hbar/m^*$, where e is the charge of the carrier, B is the magnetic field, and m^* is the effective mass of the carrier, the Landau level gaps become narrower with decreasing magnetic field B . This narrowing reduces the difficulty in determining the gaps. As a result, we focused on high filling factors at lower magnetic fields for Landau level identification. To further mitigate potential errors from manual identification, we systematically assigned the Landau levels using the standard equation $B = nh/ev$ mentioned above, where n is the carrier density, h is Planck's constant, e is the electron charge, and ν is the filling factor. This systematic approach minimizes subjective bias and enhances the accuracy and reliability of our Landau level labeling. We have included these clarifications and additional discussions in the revised manuscript to address the reviewer's concern and reinforce the robustness of our analysis.

Moreover, if the local doping at different regions of graphene flake are different, the Landau Fan diagram can also show different sets of Landau levels as shown in Fig. 4d.

Response:

Thanks a lot for the Refree#2's constructive suggestion.

Actually, the two sides peaks appearing in a mirror symmetric manner with respect to the main Dirac peak, which are shown in Fig. 4d-f in the main text, can rule out the inhomogeneity of doping in the sample. Besides, to address the possibility of doping inhomogeneity leading to different sets of Landau levels, we conducted I - V measurements on the device. The I - V curves shown in Fig. R10 display excellent linearity, indicating no Fermi level misalignment caused by doping variations. This result confirms that the material's channel region exhibits uniform transport properties without significant local doping inhomogeneity (since differently doped areas, such as p-p', n-n', or n-p junctions, will yield nonlinear I - V curves).

Fig. R10. The optical and electronic transport characteristics of Sample-S23. a) The optical image of the Sample-S23 device with pin numbers labeled. b) The I - V curves for pins 3-4 and 1-8, respectively.

Regarding the potential influence of local doping variations, we carefully examined the spatial uniformity of the graphene flake by performing measurements across multiple regions (pin1-8 and pin3-4) of the sample shown in Fig. R11. These test results confirmed that the observed Landau levels are consistent and not significantly affected by local doping inhomogeneity. The corresponding figure has already been added into the supplementary information as Supplementary Figure 17.

In addition to the devices presented in our previous submission, which utilized DNA templates with an 84 nm square periodicity, we also fabricated graphene/DNA heterostructures using DNA templates with a 36 nm hexagonal lattice symmetry. Landau fan measurement was conducted on this device (Sample-S23), which was shown in Fig. 4e in the main text in the revised manuscript. And the resulting fan diagrams further confirm the DNA superlattice modulation presented in the manuscript. These findings reinforce the robustness of our conclusions regarding the influence of DNA superlattice modulation, highlighting the consistency of our results across different periodicities.

We believe these revisions, along with the additional analyses, provide a comprehensive clarification and reinforce the reliability of our DNA superlattice modulation.

Fig. R11. The Landau fan for the device Sample-S18, which incorporates DNA templates with an 84 (42) nm square lattice symmetry, is presented. a) and b) show the optical image and schematic diagram of the graphene/DNA heterostructure, respectively. c) and d) display the Landau fan diagrams measured for pin3-4 and pin6-7, respectively. The solid black lines indicate the Landau levels for the main Dirac fan, while the solid green lines represent the superlattice satellite fans.

Comment 3. *Is it possible to generate DNA superlattice with smaller periodicity so that the signatures of DNA superlattice can be better resolved?*

Response:

We completely agree with the reviewer's opinion.

A smaller DNA superlattice periodicity would result in a reduced superperiodicity between the DNA and graphene. Consequently, the carrier concentrations corresponding to the superlattice satellite peaks would be further away from the charge neutrality point, making it easier to distinguish the side peaks from the main peak. Following review's recommendation, we explored the preparation and deposition of DNA superlattices with other periodicities.

Fig. R12. The magneto-transport behavior of the graphene/DNA heterostructure with a hexagonal DNA superlattice (wavelength = 36 nm). a) AFM characterization of the 36 nm hexagonal DNA origami 2D film. b) Zoomed-in scan of the area outlined by the green dashed box in a). The scale bars in a) and b) are 100 nm. c) Optical image of the device (Sample-S23) and schematic of the measurement setup. d) Landau fan obtained from the graphene/DNA heterostructure (Sample-S23). The solid black lines indicate the Landau levels for the main Dirac fan of graphene, while the green lines represent the additional side Dirac fans originating from the hexagonal DNA superlattice. The satellite peaks are positioned at $\pm 0.145 \times 10^{12} \text{ cm}^{-2}$ away from the charge neutrality point (CNP) of graphene, as indicated by the extensions of the green lines.

Due to the limitations of biological self-assembly methods, the smallest periodic DNA structures that can currently be stably deposited onto rigid alumina substrates, while maintaining uniformity and planarity over the micron scale, exhibit a periodicity of 36 nm with hexagonal lattice symmetry. As shown in Fig. R12a-b, the atomic force microscope images show that the DNA origami 2D film has a honeycomb distribution with a 36 nm periodicity. A standard Hall-bar device (Fig. R12c, Sample-S23) was prepared and corresponding magneto-transport was studied. Fig. R12d (also cited as Fig. 4e in the main text of our revised manuscript) demonstrates the superlattice effects induced by the DNA origami. We assigned the index of Landau levels using the equation $B = nh/ev$, where n is the carrier density, h is Planck's constant, e is the electron charge, and v is the filling factor. This approach minimizes subjective bias and enhances the accuracy and reliability of our Landau level labeling. As shown in Fig. R12d, three sets of Landau fans were assigned. Among these, two symmetric Landau fans (depicted by the green lines) are positioned at $\pm 0.145 \times 10^{12} \text{ cm}^{-2}$ away from the main Dirac fan (depicted by the black lines), agreeing well with the full filling of 4 electrons per unit cell (area $A \sim 2700 \text{ nm}^2$), which agrees with the surface area of a unit cell of the studied hexagonal DNA lattice with 36 nm wavelength (Fig. R12a-b).

These additional measurements further corroborate the reliability of our conclusions. We have included a discussion of these findings in the revised manuscript to address this point.

Again, we would express our sincere appreciation to the very constructive comments from Referee #2. Stimulated by her/his valuable suggestions, we have made corrections/revisions accordingly, and our manuscript is now a lot improved. Her/his support in publication in Nature Communications will be greatly appreciated.

Reviewer #3 (Remarks to the Author):

General Comment. *Patterning graphene with a spatially periodic potential provides a powerful means to modify its electronic properties [Nat. Nanotechnol. 16, 525 (2021); Nature Nanotech 13, 566 (2018); Nat Electron 4, 116 (2021)]. In previous works, the induced spatially potentials are usually “hard” materials. In this article, Zhao et al presented a novel strategy to alter the electronic properties of graphene by “soft” material, called DNA, which serve as a moiré-like superlattice after its self-assemblies. I think this work is very interesting, which can inspire more studies on the biology nanoelectronics. While I enjoy the results of this manuscript, there are some questions that should be addressed before I can recommend its publication. I detail my concerns below.*

Response:

We sincerely appreciate the positive and encouraging feedback provided by Referee #3. Indeed, as described in the general comment by Referee #3, there are some unclear parts in our previous submission, and we have, according to Referee#3’s comments, corrected them in this following version.

In the coming parts, we address her/his comments in a point-to-point fashion, listed below.

Comment 1.

DNA periodic potential.

Formation of DNA period. DNA origami units can be self-assembled from a long DNA scaffold and hundreds of short DNA staples, and further joint together through base pairing and stacking into a square lattice with a periodicity of ~ 80 nm. May I ask this kind of assembled process is quite random or can be controlled? That is to say, the periodicity of ~ 80 nm can be further increased or decreased?

Response:

In our last submitted manuscript, we utilized self-assembled DNA origami 2D lattices composed of square DNA origami tiles (wavelength = 84 nm). In a previous report, Yue Tang *et al.* engineered two-dimensional DNA arrays with controlled periods (36, 53, 61, 85, 94 and 130 nm) and various geometries, including equilateral triangle, square, and regular hexagon.^[R5] These lattices have been successfully assembled through thermal annealing, exhibiting well-defined

^{R5} Tang, Y., Liu, H., Wang, Q. *et al.* *J. Am. Chem. Soc.* **145**, 13858(2023).

nanometer-scale precision and micrometer-scale order. However, due to their tolerance and stability during the essential rinsing and drying steps to combine with hard matter, only the 84 nm square periodic DNA superlattice is robust enough. Other lattices were more susceptible to deformation and may disassemble during the rinsing and drying process. That's why we selected the square DNA superlattice with a periodicity of approximately 80 nm in our previously submitted manuscript. Nevertheless, after several months' optimization of the fabrication and deposition processes, now, the smallest edge of the DNA superlattice that we can prepare in a lateral size of $\sim 10 \mu\text{m}$ on a hard substrate is 36 nm (hexagonal lattice, as indicated in the Supplementary Figure 15 in the updated manuscript).

Comment 2. *The profile lines in Fig.3b are not so clear, I suggest making the lines or colours a little bit thicker.*

Response:

We sincerely thank Reviewer #3 for her/his thorough examination of our manuscript and for pointing out the clarity issue in Fig. 3a and Fig. 3b. In response, we have now bolded the relevant lines in these figures in the revised manuscript to enhance the visibility of the DNA superlattice's profile and make it more distinct.

Comment 3. *Super moiré. Usually, the super moiré is formed by stacking two more patterns together, and at least have three layers of materials. So, from my understanding, there only two layers of DNA stacking together with a twist angle, so how can we call supermoiré in Fig. 3f-i?*

Response:

We sincerely appreciate the reviewer#3 for providing this suggestion. Indeed, these two layers of DNA superlattices should not be referred to as 'supermoiré'. We have now replaced the term 'super moiré' in Fig. 3f-i in the main text with 'moiré pattern' in the revised manuscript, highlighted in blue, in Page 4.

Comment 4. *How to determine the twist angle in Fig. 3n-q? It is not so clear, and I suggest that the author makes more analysis on the fast Fourier transformation (FFT) pattern shown in Fig. 3j-m.*

Response:

We appreciate very much this very constructive suggestion by Referee#3.

Indeed, the method for determining the twist angle in our previously submitted manuscript was not clearly explained. In fact, fast Fourier transforms (FFT) were applied to SEM images using the freeware “ImageJ”, rather commonly used for image processing and analysis. As shown in Fig. R13, the twisted angles in Fig. 3f-i in the revised manuscript can be calibrated as 7.5° , 9.5° , 48.0° , and 66.5° , respectively. We have added a description of FFT processes in the revised manuscript and included the corresponding angle identification (Supplementary Figure 8) in the updated supplementary materials.

Fig. R13. The angle identification of twisted double-layer DNA tessellations through fast Fourier transformation (FFT). In a) to d), the two sets of yellow lines correspond to the upper and lower periodic DNA tessellations. The angle between these two sets of lines represents the twisted angle between the two DNA layers.

Comment 5. The reproducibility of such DNA origami induced additional Landau fan, we have collected data from another device sample 15 (S15), which shows the similar electronic behaviour. I am very impressive that the authors have studied so many devices, I think it should be more than 15 devices. I am confused the main data of Fig. 4 come from which device? It seems that there are only two devices, not 15 devices. So how about the information of other devices? The author need make this clear.

Response:

We thank very much for the reviewer’s attention to the additional Landau fan, which is also one of our key findings. First, the main data of Fig. 4 in our previously submitted manuscript is from

Sample-S18, and we have now included this device number in both the figure caption and the main text. Prior to successfully observing additional Landau fans in Sample-S15 and Sample-S18, we made extensive efforts in device fabrication, in terms of optimizing both the dielectric layer and the device structure. We have sequentially numbered all these earlier devices, which unfortunately were not able to survive till the end, as many of them were either not working or damaged during the test processes. Below, we describe this challenging process from the perspectives of dielectric layer and device structure.

1. **Dielectric layer:** The initial devices used 280 - 300 nm thick SiO₂ as the gate dielectric layer. From our analysis, the inability to observe additional Landau fans in these silicon-gated devices may be attributed to the excessive thickness of the SiO₂ layer. Subsequently, we fabricated a batch of devices using thinner mica (10 - 20 nm) as the gate dielectric layer. The mica was obtained through mechanical exfoliation, which introduced considerable randomness in the DNA superlattices deposition process. Additionally, we observed that thin mica was susceptible to leakage, likely due to defects introduced during the mechanical exfoliation process. Ultimately, we resolved these issues concerning the excessive silicon gate thickness and leakage in mechanically exfoliated mica by utilizing a 15 nm-thick Al₂O₃ (or HfO₂) dielectric layer deposited through atomic layer deposition.

2. **Device structure:** Initially, we employed a device structure in which graphene was in direct contact with the DNA superlattice. This resulted in low graphene mobility and extremely poor quantization. Subsequently, by adopting an h-BN-encapsulated graphene structure with the bottom h-BN layer about 1 nm thick, the mobility and device performance were significantly improved. Meanwhile, some devices exhibited alignment between h-BN and graphene, hindering our analysis of the satellite peaks modulated by DNA superlattice.

In summary, we fabricated a considerable number of devices with varying dielectric layers and structures. These devices were all traced with numeric labels to enable a systematic analysis.

Origin of mini-Landau fan diagram.

Comment 6. *Fig. 4 is a little confused. In order to show the effect of DNA on the electronic property of graphene, the optical device of graphene without DNA and with DNA is shown in Fig. 4b, but the Fan diagram is just showing the graphene device with DNA (Fig.4d). It is better to show the fan diagram of graphene with and without DNA in the Fig. 4, like Fig.3 in Ref. [35]. If the author wants to further to claim the effect of DNA, similar with the effect of aligned G/hBN. Then we can also put the third maps in Fig. 4. For example, Graphene without DNA, Graphene with DNA and Graphene aligned hBN.*

Response:

We sincerely appreciate Reviewer #3 for her/his constructive suggestion regarding the layout of Fig. 4 in our previously submitted manuscript.

Indeed, arranging three Landau fan diagrams vertically for comparison makes it much easier to observe the effect of DNA superlattice on the electronic property of graphene. After preparing the new graphene/DNA device with hexagonal DNA superlattice (wavelength = 36 nm), we arranged three Landau fan diagrams vertically, which are as follows: graphene with square DNA superlattice (wavelength = 84 nm), graphene with hexagonal DNA superlattice (wavelength = 36 nm), graphene with square DNA superlattice (wavelength = 84 nm) and h-BN/graphene aligned. It has been noted that we have newly assigned the Landau levels using the standard equation $B = nh/ev$, where n is the carrier density, h is Planck's constant, e is the electron charge, and v is the filling factor. This systematic approach minimizes subjective bias and enhances the accuracy and reliability of our Landau level labeling. These linear feature lines ($B \sim n$) within the Landau gaps at different filling fractions have been extracted and plotted together in Fig. 4g in the revised manuscript. It is clearly observed that these feature lines are generated from graphene (black lines), DNA superlattices (red and green lines for square and hexagonal DNA superlattices, respectively), and h-BN/graphene alignment induced moiré superlattice (light blue lines). However, due to the limitation of the figure size, we show it in the supplementary information, as Landau levels for plain graphene has been reported widely in literatures. Therefore, we beg to put them in the supplementary information instead of adding it into the main figure in our revised manuscript. And we thank very much for Referee#3's understandings and supports.

Fig. R14. (Cited as Fig. 4 in revised manuscript) A 2D interface of soft- and hard-matter. a) SEM image of the DNA origami 2D film on top of the pre-patterned bottom Au gate covered by 15 nm Al_2O_3 . The DNA 2D film is outlined by the white dashed line. b) Device fabrication of a typical vertical heterostructure of BN (20-30 nm)/graphene/BN(1-3L) on top of the DNA film in a). The electrodes are patterned using standard lithography followed by metallization. Hall bars of the heterostructure are patterned via plasma etching. c) Cartoon illustration of the vdW heterostructure. d) Landau fan map ($\log(R_{xx})$) recorded in the parameter space of magnetic field B and carrier density n of a typical device with DNA superlattice (Sample-S18) as shown in b). f) Control sample of a DNA/h-BN/graphene/h-BN heterostructure with an alignment angle close to zero degree. Samples in d) and e) are intentionally made to avoid alignment of graphene and h-BN during device fabrications. g) The Landau Levels (LLs) (Black: graphene; Red: square DNA superlattice; Green: hexagonal DNA superlattice) extracted from d) and e), with the filling fractions labelled in each LL. The blue dashed line in f) and g) indicates the minimum carrier density for the origin of a Landau fan induced by almost perfectly aligned graphene/h-BN in our control sample, which is much higher than the observed mini-fan (black solid lines) in d).

Comment 7. *To demonstrate the period moiré potential effect, there are two effects: One is the second Dirac point, and the second the BZ oscillations which can survive high temperature. These two effects have been observed in precious moiré potential paper [Nat Commun 14, 4142*

(2023)]. I suggest the authors cite this paper, and discuss why in current graphene/DNA system, there is no signature of Brown-Zak features.

Response:

We thank the reviewer for highlighting the Brown-Zak oscillations (BZO) as a potential signature of the moiré potential and for suggesting the reference [Nat Commun 14, 4142 (2023)].

However, we do not have the fortune to observe the BZO in our devices. We speculate that it may come from the relatively weak interaction between graphene and the DNA superlattice, due to the modulation of spatial distribution of electrical fields thanks to the only 1–2 nm thickness of the DNA networks. On the contrary, in systems such as twisted bilayer graphene or graphene aligned with hexagonal boron nitride (*h*-BN), those system are subjected to a very strong atomic potential that yields significant hopping of electrons in the moiré sites, instead of gate-induced electrical field modulation in our case (please see the COMSOL simulations in Supplementary Figure 11). This weaker coupling likely contributes to the absence of detectable BZO.

We have cited this paper in the revised manuscript and added a discussion highlighted in blue in Page 6, and we quote it as follows:

“Brown-Zak oscillations (BZO) is found to be also evidence for superlattice effects in some systems ⁴¹. However, we regret that in our current study, no such BZO were seen, which may be due to the relatively weak interaction between graphene and the DNA superlattice.”

We appreciate the very important comments given by Referee#3 here.

Comment 8. *The band structure of graphene can be reconstructed due to the existence of an artificial superlattice, and the DNA will change the band structure of graphene. So how the band structure will be changed? Such a modulation of periodic potential is able to yield effective band reconstructions in graphene. I suggest the authors need discuss a little about it at the end.*

Response:

We appreciate the reviewer’s insightful comment regarding the band structure reconstruction in graphene due to the periodic potential of the DNA superlattice. This is indeed an important aspect of our study. The DNA superlattice introduces an artificial periodic potential to the graphene layer, which modulates the electronic states and leads to the formation of new mini-bands. Such periodic potentials can reconstruct the original linear band structure of graphene by creation of mini-

Brillouin zones, generating additional minima in DOS at slightly higher energy and thus creating additional gaps at certain high-symmetry points in the reciprocal space.

In our study, the DNA templates with periodicities of 36 nm or 84 nm impose a weak moiré potential on graphene due to the spatial modulation of electrical fields (as supported by COMSOL simulations in Supplementary Figure 11). This modulation is sufficient to observe secondary resistive peaks (i.e., weak gaps) and Landau level splitting, as shown in the Landau fan diagrams. These features indicate the formation of new band structures induced by the periodic DNA superlattice.

Other examples of such superlattice modulation of band structure in graphene have also been reported in literatures, including graphene placed above the holed arrays of etched SiO₂ substrate,^[R6] h-BN/Graphene/h-BN etched through into an ordered array,^[R7] or a square lattice etched using focused ion beam etched template.^[R8]

However, due to the relatively large periodicities and possible interfacial disorder, the modulation strength in our current study may not be as pronounced as in systems like twisted bilayer graphene, where strong moiré potentials cause more dramatic band reconstructions. We have added a schematic drawing in Supplementary Figure 9 in the updated manuscript, in order to explain how the periodic potential may influence the electronic dispersion/band-structure. We quote it below:

Fig. R2. (Cited as Supplementary Figure 9 in the updated Suppl. Info.) Comparison of trivial charge-impurity doping and the superlattice effect on graphene. a) Trivial charge-impurity doping and the consequence of Fermi level shifting of graphene without band reconstruction. b) Band reconstruction of graphene due to the existence of an artificial superlattice. Two side peaks can be seen due to the formation

^{R6} Forsythe, C., Zhou, X., Watanabe, K. *et al. Nat. Nanotechnol.* **13**, 566–571 (2018).

^{R7} Jessen, B., Gammelgaard, L., Thomsen, M. *et al. Nat. Nanotechnol.* **14**, 340 (2019).

^{R8} Ruiz, D., Sheinfx, H., Hoffmann, R. *et al. Nat. Commun.* **13**, 6926 (2022).

of additional minima in the DOS (which yield a resistance maximum accordingly, in the transport behaviors). For example, it can be realized by etching the SiO₂ substrate into holed arrays with a pitch of ~ 30 nm.^[R⁹]

Finally, we feel very grateful that Referee#3 have provided very thoughtful and helpful recommendations, that have led to a number of corrections/revisions in this new submission. Thanks to her/his valuable suggestions, we were able to better our manuscript significantly, and her/his support in publication in Nature Communications will be very much appreciated.

^{R⁹} Forsythe, C., Zhou, X., Watanabe, K. *et al. Nat. Nanotechnol.* **13**, 566–571 (2018).

Rebuttal Letter

Reviewer #1 (Remarks to the Author):

General Comment. *The revisions by Zhao et al. corrected few grammar and reference mistakes in the initial submission. However, unlike their statements in the rebuttal letters, the authors did not address the key technical concerns raised by the reviewer, i.e. the physical mechanism in Landau fan map. The proposed “DNA superlattice effect” still lacks enough experimental support, and was not consistent with the newly supplemented results, which are in line with the widely reported temperature effect, ion effect, and surface impurity effect. More importantly, the reviewer noticed that several figures of the work were likely re-used, labelled with different conditions, which raised the concerns over the data integrity. Based on these concerns, the reviewer strongly recommends to reject this manuscript.*

Response:

We thank the comments given by Referee#1 in this new round of review.

However, we found his comments invalid. Below, we explain why in the point-by-point response.

Comment 1. *1. In Supplementary Figure 4, it is clear that Figure 4a (claimed to be after soaking in ethanol), Figure 4b (claimed to be after soaking in isopropanol solution), and Figure 4d (claimed to be after soaking in trichloromethane solution) have identical imaging details, noise level and sample defects, which are unlikely reproducible in different measurements. Similar issues could also be found in Supplementary Figure 4c and Figure 4e, where different processing conditions have been used. It seems that Supplementary Figure 4a, Figure 4d and Figure 4e are only parts of Figure 4b and 4c, with modified contrasts, height axis, and image dimensions, but labelled with totally different conditions.*

Another figure similarity occurs between Figure 4d in the main text (also Supplementary Figure 14a) and Supplementary Figure 17 c, where the authors claimed measured from different pin regions in Sample-S18. Despite different color settings and x-axis ranges were used in these figures, all the other figure details, including noise level and distribution, are identical. If all these figures are from one device, why the authors report the same figure three times?

Response:

1. “... have identical imaging details, noise level and sample defects, which are unlikely reproducible in different measurements ...”.

-This is NOT true.

We would like to remind the Referee that, what is shown in Supplementary Figure 4, is a “SEQUENTIAL” soaking of the SAME piece of DNA film in solutions of ethanol, isopropanol, acetone, and trichloromethane.

By using the same DNA film, we wanted to show which specific immersion process within the standard nano-fabrication procedure might potentially harm the DNA film. And, remarkably, our results revealed that the DNA film did not display significant disassembly or wrinkling phenomena following immersion in various organic solutions and subsequent vacuum baking.

2. *“Supplementary Figure 4a, Figure 4d and Figure 4e are only parts of Figure 4b and 4c, with modified contrasts, height axis, and image dimensions, but labelled with totally different conditions”.*

-This is NOT true.

The Referee is simply lying and being irresponsible to his/her words. There is no modifying of contrast, nor height axis, nor dimensions. These figures are re-captured by AFM after each processing. -- They are literally different measurements.

3. *“several figures of the work were likely re-used, labelled with different conditions, which raised the concerns over the data integrity”*

-This is NOT true.

It is absolutely normal to show the same figure as the main text in the Suppl. Info., just in order to discuss several different question, and make it more convenient for the readers to have a direct comparison when reading.

In fact, we have made it crystal clear that these figures are indeed derived from the same device (Sample-S18) in the figure captions in both the main text and the supplementary information. Specifically, Figure 4d, Supplementary Figure 14a, and Supplementary Figure 17c represent the test result from the same pair of pins. These arrangements were established to facilitate direct comparisons:

- Supplementary Figure 14a was placed alongside Supplementary Figure 14b to highlight the differences between the Landau fans with and without the DNA superlattice.

- Supplementary Figure 17c and Supplementary Figure 17d were shown together to demonstrate that testing different pins within the same Hall bar device both revealed additional Landau fan features.

Therefore, **the Referee#1’s accuses of “figures of the work were likely re-used, labelled with different conditions, which raised the concerns over the data integrity” are exaggerated and are invalid.** We feel sorry and found it unfair that Referee#1 drew extremely biased conclusions on our manuscript, without reading it carefully.

Comment 2. *The core conclusion of this work, as argued by the authors, is the “superlattice effect” of DNA structures on 2D materials. To support this conclusion, two experimental results are necessary, including the reproducible device construction to exclude random effect in sample preparation and the same-condition measurements of DNA samples with different periodicities/heights/surface morphologies to verify the effect arising from DNA molecules (even without superlattice) or the periodicities of DNA superlattice. The authors did provide a new device in Figure 4e. However, this new device (measured at 25mK) is not measured at identical condition to the other device (1.5 K), and cannot be used for direct comparison to support their conclusion on superlattice effect. Temperature effect plays a more important role here. The temperature effect could not be excluded here, and it combined with other effects discussed below. Therefore, it still lacks experimental evidence to confirm the superlattice effect.*

Response:

We thank the reviewer for raising concerns about the temperature difference between the measurements of the devices in Figure 4e (25 mK) and the other devices (1.5 K). We would like to emphasize that this temperature difference is not a critical factor in our experiments and does not directly affect our conclusions or the solid, reproducible characteristics of the samples.

In our measurements, we are measuring graphene, and it is spaced by a ~ 1 nm h-BN layer (h-BN is an insulator) between the underneath DNA origami tessellation. There is absolutely no such “temperature effect” or what so ever.

While lower temperatures (25 mK) help reduce thermal disturbances and improve the contrast of the Landau fan, the positions of the Landau fan features remain unaffected by temperature. This ensures that the superlattice effects observed in our devices are intrinsic and independent of the temperature differences in the measurements.

Comment 3. *The superlattice morphologies of DNA on real device samples are missing. Regarding the AFM images in Fig R8 and Supplementary Figure 15, these small-scale images (around 200 nm) do not reflect the lattice uniformity for the whole device (longer than 5 μ m). Considering their DNA lattices adopt different morphologies and periodicities on different substrates, the true morphology on device is important. However, the authors did not indicate where the sample was measured. Considering DNA structures may be distorted with uncontrollable lattice morphology, without the high-resolution true morphology on a real device surface, it would be difficult to conclude if there was DNA superlattice effect.*

Response:

We appreciate the reviewer's attention to the DNA superlattice's morphology on the final device.

1. “*these small-scale images (around 200 nm) do not reflect the lattice uniformity for the whole device (longer than 5 μm)*”.

- **This is NOT true.** We have large scale AFM/SEM images (over 10 μm) shown in Figure 3a in our main text, which speaks the lattice uniformity of those DNA origami tessellations.

2. “*However, the authors did not indicate where the sample was measured.*”

- **This is NOT true.**

It is clearly marked in the figures where the electrodes were fabricated. We use standard 4-probe measurements to measure the graphene channel between the two voltage electrodes, with current injection and ground connected to the other two electrodes.

3. “*DNA structures may be distorted ..., without the high-resolution true morphology on a real device surface...*”

- In our final device, a van der Waals heterostructure (BN/graphene/BN) were deposited on top of the DNA superlattice, it is **technically impossible to see through** the whole heterostructure and to probe the interface with high resolutions.

The reviewer raised concerns regarding the potential degradation of DNA during our micro-nanofabrication processes. This is consistent with the focus of our study on investigating the compatibility of micro-nanofabrication techniques with DNA integrity. As shown in Figure 2g and Supplementary Figure 4, the entire fabrication process preserves the integrity of the DNA superlattice, even in the presence of organic solvents and heating effects.

Comment 4. *More importantly, as shown in Figure 4a, the sample surface was covered with many assembly impurities (shown as the dark regions in the device areas in Figure 4b), which apparently affected the Landau fan diagram (as shown in Supplementary Figure 17). The same surface impurity would also appear, and deviate the Landau fan even when different DNA periodicities are used. The authors did not analyze these impurity effect, and ascribe the effects solely to DNA superlattice, which is not correct.*

Response:

We acknowledge the reviewer’s observation regarding surface impurities and totally agree that our sample surfaces are not perfect, as is already well noted in the manuscript at the very beginning – **we never claimed that we have a “perfect” DNA superlattice.**

However, since these impurities are isolated and do not form a periodic structure, they do not impact the core observation of the additional symmetric set of Landau levels induced by the DNA superlattice.

Long range order, is something that Referee#1 has missed, and is the key to understanding the solid state physics reported here (and in many superlattice induced bandstructure engineerings in our previous rebuttal letter, which we will not repeat here in this letter).

As a good reference, in twisted 2D material systems such as twisted bilayer graphene, large-scale periodic inhomogeneities can exist due to interface strain relaxation or other factors. Despite these inhomogeneities, novel quantum phenomena such as superconductivity and topological correlated states have still been observed.

Comment 5. The authors argued that ion traps would lead to large hysteresis. However, this is not true. Considering the thin thickness of DNA, short-range movements (less than 1 nm) of ions did not introduce large hysteresis, this is different from those in bulk dielectric. In addition, discussion on DNA origami in electric field has been reported before (Li et al., ACS Nano 2015). Therefore, the simulation in Supplementary Figure 11 is technically wrong, because its model did not reflect the ionic nature of DNA. Besides, even though h-BN is used to block ions, these moving ions could still affect graphene transport, by generating strong electric field near graphene/h-BN surface, and did not need to be trapped in graphene lattice. Thus, the authors did not have enough evidence to exclude ion effect.

Response:

We thank the reviewer for insisting his/her picture of “ionic nature” of DNA.

1. *“discussion on DNA origami in electric field has been reported before (Li et al., ACS Nano 2015)”*.

-We believe that the Referee#1 was talking about Li et al, ACS Nano, 9, 1420 (2015). However, **that is a molecular dynamics simulation work, which investigates DNA in liquid.** – Which has nothing to do with the solid state devices in our case. One may talk about ions in liquid, but in our work, the DNA origami is assembled from insulating short segments/strands, and the superlattice is deposited onto wafers, dry.

2. *“the simulation in Supplementary Figure 11 is technically wrong, because its model did not reflect the ionic nature of DNA”*.

-The Referee#1 has made an invalid point here.

A). His/her minds are very much limited to the literature on liquid phase studies of long-chained DNA with ions movable in them, and direct measure of the conductivity of DNA.

B). However, in our case, we use insulating, short, DNA strands, and we measure indirect effects caused by the long-range orderings formed in the DNA 2D superlattice.

These two approaches (A & B mentioned above) of DNA investigations, are totally different areas. They are simply not comparable.

3. “*even though h-BN is used to block ions, these moving ions could still affect graphene transport*”.
-**This is NOT true.**

Please refer to our response to Comment 6 below.

4. “*by generating strong electric field near graphene/h-BN surface, and did not need to be trapped in graphene lattice*”.

- **This is NOT true.**

In fact, if there were indeed such movable ions, they would screen the back gate, and they would never be ordered. – Nevertheless, we do not have such “moving” ions in our system, as explained in detail in our response to Comment 6 below.

First part of Comment 6. *The authors believed that DNA superlattice effect was their key discovery. But such effect lacks supports, and even is contrary to the results. As shown in their experiments and discussed above, the authors did not consider the temperature effect, the impurity effect and ion effect, and did not provide enough evidence showing the large-area DNA morphology on real device surface. In addition, the key hypothesis of DNA superlattice effect was DNA structures as dielectrics (as shown in Figure 1e), which are directly contrary to previous reports using DNA as conductors (Fink et al., Nature 1999) and superconductors (Yu et al., Science 2001) at such low temperatures.*

Response:

The reviewer raised concerns about the potential contradiction between our hypothesis that DNA structures act as dielectrics and previous reports of DNA exhibiting conductive or superconductive behavior under specific conditions (e.g., Fink et al., Nature 1999; Yu et al., Science 2001).

However, **none of the two papers mentioned by Referee#1 is relevant** to the current study.

As shown in Fig. R1, these studies on DNA conductivity typically involve placing double-stranded DNA between electrodes. This approach completely differs from our system. Additionally, the integrity of the DNA molecule and the presence of breaks in the double helix significantly influence conductivity. The two referenced studies use intact, long double-stranded DNA, while our DNA origami structure is assembled from multiple short DNA strands, making direct comparisons impractical.

[REDACTED]

Fig. R1 DNA conductivity Testing along the double double-stranded DNA. a) Schematic of low-energy electron point source (LEEPS) microscope used to investigate the conductivity of DNA molecules. ^[R1] b) Schematic drawing of the measured sample, with DNA molecules combed between Re/C electrodes on a mica substrate. ^[R2]

Fig. R2. The role of the backbone in the I - V characteristics. ^[R3] When there is a nick point on the double-stranded DNA, the conductivity performance decreases, indicating the crucial role of the continuity and integrity of the DNA double helix in its charge transport.

The conductive properties of DNA are multifaceted, with studies reporting behaviors ranging from insulating to conductive under different setups and conditions. Whether DNA molecules demonstrate conductive or insulating characteristics is conditional and determined by factors such as molecular orientation, sequence, strand continuity, and complementarity. Generally, DNA can exhibit conductivity along the direction of the stacked base pairs (i.e., the helical axis), as the π -electrons in the nucleobases facilitate charge transport. Charge transport in DNA is dominated by

^{R1} Fink, H., Schönemberger, C. *Nature* **398**, 407-410 (1999).

^{R2} Kasumov, A., Kociak, M., Gueron S., et al. *Science* **291**, 280-282 (2001).

^{R3} Bae, S., Lu, K., Han, Y., et al. *Nat. Nanotechnol.* **15**, 272-276 (2020).

guanine because its highest occupied molecular orbital (HOMO) level is closest to the electrode Fermi level (Chem. Phys. Lett. 1975, 34, 11). As shown in Fig. R2, conductivity requires an uninterrupted path for charge transfer, and any structural discontinuities, such as nicks, gaps, or breaks in the sugar-phosphate backbone, can severely impede charge flow.

In contrast to double-stranded DNA, single-stranded DNA lacks the stable π - π stacking interactions necessary for efficient conductivity (Fig. R3).^[R4] The factors facilitating DNA's conductivity, as discussed above, are absent in our system.

- 1) In our system, the DNA double helices are oriented in parallel to the electrodes, rather than perpendicular, which disrupts the alignment necessary for efficient charge transport along the stacked base pairs.
- 2) The DNA sequences utilized in our origami structures contain a low proportion of consecutive cytosine-guanine (CG) base pairs, which are essential for facilitating effective π - π stacking interactions and charge transfer.
- 3) Each individual DNA origami structure in our system contains hundreds of nick points along the sugar-phosphate backbone. These discontinuities interrupt the continuity of charge transport pathways, impeding conductivity.
- 4) The double-stranded domains in our structure are interconnected by single-stranded linkers. These single-stranded regions lack the stable π - π stacking interactions necessary for efficient charge transport.

[REDACTED]

Fig. R3. The Figure adopted from Ref. [R4], which shows that the double-stranded DNA (dsDNA) is 30-fold more conductive than single-stranded DNA (ssDNA).

In summary, the structural and sequence-specific features of our DNA system, namely the parallel orientation to electrodes, low CG content, numerous nick points, and single-stranded linkers, collectively render it unsuitable for charge transport, supporting its role as a dielectric material rather than a conductor.

^{R4} Gupta, N., Wilkinson, E. A., Karuppanan, S., et al. *J. Am. Chem. Soc.* **143**, 20309-20319 (2021).

Second part of Comment 6. *The physical basis of this work is not solid. More importantly, the authors were using a geometry periodicity (30-40 nm), rather than the real sequence periodicity (100-200 nm) to fit their data. This is direct contrary to DNA superlattice effect. Because each nucleotide has distinct charge binding capability and surface electric field diffusion, their potential distribution could not be the same if their underlying sequences are different. As results, when DNA superlattice effect applies, the periodic potential will be sequence-determined, rather than morphology-determined (as used by the authors). So the periodicities used by the authors are just denying the proposed superlattice effect, and supporting a ion effect (which is consistent with geometric periodicity) mixed with impurity effect (consistent with Supplementary Figure 17).*

Response:

As we have explained in the response to the First part of Comment 6, there is no ions in our DNA strands, as they are dielectric and has no ionic nature.

To summarize, the superlattice in the current solid state device stems from in the morphology, as evidenced by AFM/SEM, not from the “ions” (which does not exist) in the 100-200 nm original short-strand DNA sequence.

Finally, we have to emphasize once more: **Long range ordering is key to understand the solid state physics reported here.** The referee has no valid point to attribute the physics to a moving-ion picture – not to mention that such moving ions do not exist in our system.

Reviewer #2 (Remarks to the Author):

General Comment. The authors have addressed all of my questions/comments. I recommend its publication.

Response:

We would like to express our gratitude once again for all the professional suggestions and positive feedback from Referee#2. His/her support of publication is very much appreciated.

Reviewer #3 (Remarks to the Author):

General Comment. *I have carefully reviewed the rebuttal letter, and the authors have thoroughly addressed my concerns. The additional data on DNA origami deposition and heterostructure devices with a new-pitch DNA superlattice significantly strengthens the work. The authors have overcome major technical challenges, like integrating DNA deposition, van der Waals material transfer, stacking on DNA origami, and nanoscale device fabrication into a unified process—a big achievement. To my best knowledge, the combination of 2D materials with large-scale DNA origami and the demonstrated superlattice effect (graphene band structure engineering via DNA-induced super-potential) marks a groundbreaking advancement in DNA-based nanoelectronics with great potential. I fully support the publication of this work in Nature Communications.*

Response:

We sincerely appreciate the positive feedback from Referee#3 and are grateful for the constructive suggestions, which have been invaluable in enhancing the quality of our manuscript.

Rebuttal Letter

Reviewer #4 (Remarks to the Author):

General Comment. *The study by Zhao et al. on the superlattice effect using DNA templates has significant academic value in two major aspects: 1) It enables the fabrication of high-quality nanostructures over a large area through a remarkably simple process, and 2) by designing the nanostructures as desired, it allows modulation of the quantum properties of 2D electronic devices. The authors have put in extensive experimental efforts and actively addressed reviewers' concerns, ultimately improved the manuscript to a level suitable for Nature Communications. I would like to contribute to enhancing the quality of this paper by providing a few minor comments shown below:*

Response:

We appreciate very much the reviewer for her/his positive and encouraging comments on our work. We are grateful that Referee #4 valued our fabrication of DNA nanostructures and the modulation of quantum properties of 2D electronic devices as “*The authors have put in extensive experimental efforts and actively addressed reviewers' concerns, ultimately improved the manuscript to a level suitable for Nature Communications*”.

Below, we provide detailed responses to each of the reviewer's comments in a point-by-point format.

Comment 1. *Figure 2a presents a schematic of the tessellation pattern formed using DNA origami units. However, the explanation of the principle behind the formation of this tessellation pattern is highly insufficient. To understand the formation mechanism, readers are required to refer to ref. 33, which is inconvenient from the reader's perspective. Instead of describing the fabrication method, the paper should provide the fundamental principles or design rules governing the formation of the tessellation pattern.*

Response:

We are very grateful to the reviewer for this critical suggestion. Indeed, re-directing the readers to Ref. [33] (the previous reference index) is very inconvenient, and a detailed description of the formation of the tessellation pattern will be helpful in terms of improving the current manuscript.

According to the referee's suggestion, we have explicitly explained, in the revised manuscript, the fundamental principles of the DNA origami technique, as well as the mechanisms and interactions

that enable the assembly of DNA origami nanostructures into 2D tessellation patterns. We also provide a schematic illustrating how the square DNA origami nanostructure with tessellation capability is designed from scratch (Fig. R1, see also Supplementary Figure 1), which culminates in the generation of staple sequences for synthesis. This bridges the information gap preceding the fabrication method, ensuring a seamless transition between design and experimental implementation.

The updated explanation is provided as follows:

Synthesis and characterizations of 2D DNA films.

The DNA origami technique^[R1] is a powerful nanofabrication method that enables the precise, programmable assembly of DNA molecules into complex nanostructures. It involves the folding of a bacteriophage-derived, long single-stranded DNA (scaffold) by hundreds of short synthetic oligonucleotides (staples) into designed shapes through Watson-Crick base-pairing. Rationally designed, geometrically compatible DNA origami nanostructures can be further joined together into tessellation patterns through base pairing and stacking interactions strategically positioned at their edges. With the continuous development of design principles and strategies, various tessellation patterns of DNA origami have been reported^[R2, 3, 4, 5, 6, 7, 8, 9, 10,11,12], with a few reaching a lateral size of a few μm . We here adopted a recently reported, optimized DNA origami tessellation system in which a critical design parameter (termed the ‘interhelical distance’) determining the conformation and tessellation capability of DNA origami units was identified and properly fine-tuned to minimize undesired curvature in monomeric DNA origami units, enabling the formation of diverse crystalline-like lattices ranging from tens to hundreds of square micrometers^[R6]. A detailed design workflow of the square DNA origami nanostructure for tessellation is provided in Supplementary Figure 1. As shown in Fig. 2a, monomeric DNA origami units can be self-assembled from the DNA scaffold and staples, and further joined together into a square lattice with a periodicity of ~ 80 nm. A unit cell of such square lattice is presented in the

^{R1} Rothemund, P. W. *Nature* **440**, 297–302 (2006).

^{R2} Liu, W., Zhong, H., Wang, R., *et al.* *Angew. Chem. Int. Ed.* **50**, 264–267 (2011).

^{R3} Liu, W., Halverson, J., Tian, Y., *et al.* *Nat. Chem.* **8**, 867–873 (2016).

^{R4} Tikhomirov, G., Petersen, P., *et al.* *Nat. Nanotechnol.* **12**, 251–259 (2017).

^{R5} Tikhomirov, G., Petersen, P., *et al.* *J. Am. Chem. Soc.* **140**, 17361–17364 (2018).

^{R6} Tang, Y., Liu, H., Wang, Q., *et al.* *J. Am. Chem. Soc.* **145**, 13858–13868 (2023).

^{R7} Wang, P., Gaitanaros, S., Lee, S., *et al.* *J. Am. Chem. Soc.* **138**, 7733–7740 (2016).

^{R8} Aghebat Rafat, A., Sagredo, S., Thalhammer, *et al.* *Nat. Chem.* **12**, 852–859 (2020).

^{R9} Wang, X., Jun, H. & Bathe, M. *J. Am. Chem. Soc.* **144**, 4403–4409 (2022).

^{R10} Chen, C., Luo, X., Kaplan, A. E., *et al.* *Sci. Adv.* **9**, eadh8508 (2023).

^{R11} Liu, Y., Dai, Z., Xie, X., *et al.* *J. Am. Chem. Soc.* **146**, 5461–5469 (2024).

^{R12} Hayakawa, D., Videbæk, T. E., Grason, *et al.* *ACS nano* **18**, 19169–19178 (2024).

solid black box, with the DNA joints highlighted in the purple circle inset in the top-right panel of Fig. 2a.

The above paragraph has already been updated in the revised maintext, in pages 2-3, highlighted in blue.

Fig. R1 Schematic illustration of the design workflow of the square DNA origami nanostructure for tessellation. (a) The square DNA origami nanostructure is composed of four copies of a repeating subunit equivalent to an isosceles right triangle with a 90° vertex angle (α). (b) The subunit can be depicted as an even number (22, in this case) of regularly-spaced line segments perpendicularly aligned at the edge opposite to α . Each line segment is a simplified representation of a DNA double helix. Inset: Two key design parameters, interhelical distance (D) and helical extension (E), are defined to describe the distance between neighboring DNA helices and their difference in length, respectively. (c) Line segments are converted into double-stranded DNA in the Euclidean space based on the structural attributes of B-form DNA. The $(0, 0, 0)$ point is assigned to the vertex of the subunit. The strands for creating the scaffold are purple-colored, while their complementary strands for creating staples are gray-colored. The two terminal nucleotides of neighboring purple strands are connected by single-stranded linkages (termed scaffold loops). Inset: Illustration of an example scaffold loop connecting the purple strands in the second and third helices. (d) Four copies of the subunit are joined together to obtain the prototypic origami nanostructure, in which neighboring subunits are bridged by scaffold and staple bridges (inset). The design is finalized using Tiamat, a design software tool for designing DNA nanostructures from scratch, with sequence of the scaffold strand assigned. The matching rule governing the interaction between monomeric DNA origami nanostructures is defined by prescribing base pairing and stacking interactions at their edges (inset). Sequences of the staple strands are generated as the output for synthesis (listed in the Supplementary Tables 1-5).

The above figure is also updated in the revised Supplementary Information, in page 3, highlighted in blue.

Comment 2. *Supplementary Figures 2–4 show the formation of DNA 2D films on various substrates. However, there is no discussion on why these experiments were conducted or what their results imply. First, the rationale behind substrate selection is missing—what differences were the authors aiming to observe? Additionally, while the results are presented, there is no discussion or conclusion regarding their implications. Were these experiments conducted to demonstrate thickness tunability? Or to show variations in the periodicity of the tessellation pattern? If the modulation of DNA films is influenced by the substrate, what is the underlying mechanism behind this phenomenon?*

Response:

We appreciate the reviewer’s insightful comment.

It’s very true that a description part of the observations in Supplementary Figures 2-4 were missing, which is now amended in the revised manuscript. We thank the referee for her/his very helpful comments.

In the revised Supplementary Information, pages 7-8, we have added the following text, highlighted in blue:

“In the community of biosynthesis, DNA origami assemblies are often deposited on mica and characterized mostly in liquid phases, which is very different when in the cases of solid state nanoelectronic devices. Here, in our work, the primary objective of presenting Supplementary Figures 2–4 (now indexed as Supplementary Figures 3-5) is to demonstrate the compatibility of soft matter DNA with various hard substrates, as well as the solvents used in nanofabrication processes – since the solid states nanoelectronics are usually built on typical substrates such as silicon oxides. To achieve this, we deposited 2D DNA origami flakes from solution onto different substrates, followed by drying, and examined its morphology and periodicity using Optical Microscopy (OM), Atomic Force Microscopy (AFM), and Scanning Electron Microscopy (SEM).

By evaluating how well the intrinsic periodicity of DNA was maintained on various substrates—including h-BN, graphene, SiO₂ (300 nm)/Si⁺⁺ wafer, Al₂O₃, HfO₂, and mica (Supplementary Figures 2–3, now indexed as Supplementary Figures 3-4 in the revised manuscript files)—we identified Al₂O₃ as the most suitable substrate for subsequent heterostructure fabrications. Notably, Al₂O₃ deposited via the atomic layer deposition method provides a large-scale, uniform, and insulating film, making it an ideal choice for preserving DNA’s periodic structure.

Additionally, we tested the stability of DNA’s periodicity by immersing it in different solvents commonly used in device fabrication (Supplementary Figure 5 in the updated submission). These experiments confirmed that the solvents do not cause severe damages to DNA’s periodic structure,

ensuring its compatibility with the following up nanofabrication processes. These findings are crucial in selecting appropriate substrates and solvents that preserve the DNA superlattice, providing references for possible integration into heterostructures with hard materials like graphene.”

Comment 3. *Figures 2b and 2c show OM images and color intensity of the 2D DNA film. However, the significance of these data is unclear. As presented, these figures merely indicate color variations in different regions, rather than demonstrating the presence of DNA. For example, in the case of graphene, the light-matter interaction results in an absorption of ~2–3% across most spectral ranges, allowing monolayer graphene to be identified via optical contrast in an optical microscope. Similarly, the paper should explain the mechanism by which the DNA superlattice interacts with light in the relevant spectral range, the extent of absorption, and how this leads to the observed optical contrast changes in OM images. Without this information, these figures appear unnecessary in the main text.*

Response:

We thank very much this great point raised by Referee#4.

To explain the observed variations in light reflection across different wavelength, we follow the approach outlined in published paper^[R13]. We consider the case of normal incidence of light from air (with refractive index $n_0 = 1$) onto a trilayer structure consisting of DNA, SiO₂, and Si. The Si layer is assumed to be semi-infinite and characterized by a complex refractive index n_3 , which is wavelength (λ) dependent^[R14]. The SiO₂ layer has a thickness $d_2 = 300$ nm and a refractive index n_2 , which is purely real and also wavelength(λ) dependent^[R15, 16]. The DNA layer, with thickness d_1 , is considered to be 2 nm, as shown in Figure 3c of our main text. The complex refractive index n_1 of DNA is treated as a fitting parameter in our calculations.

With this configuration, we derive the reflection coefficients and the reflected intensity.

The reflection coefficients are given by:

$$r_1 = \frac{n_0 - n_1}{n_0 + n_1} \quad (1)$$

$$r_2 = \frac{n_1 - n_2}{n_1 + n_2} \quad (2)$$

^{R13} Blake, P., Hill, E.W., Castro Neto, *et al.* 2007. *Appl. Phys. Lett.* **91** (2007).

^{R14} Aspnes, D.E., Studna., A.A., *Phys. Rev. B* **27**, 985-1009 (1983).

^{R15} Malitson, Ian H. *J. Opt. Soc. Am.* **55**, 1205-1208 (1965).

^{R16} Tan, C. Z. *J. Non-Cryst. Solids* **223**, 158-163 (1998).

$$r_3 = \frac{n_2 - n_3}{n_2 + n_3} \quad (3)$$

And the phase shifts are defined as:

$$\Phi_1 = \frac{2\pi n_1 d_1}{\lambda} \quad (4)$$

$$\Phi_2 = \frac{2\pi n_2 d_2}{\lambda} \quad (5)$$

And the total reflected intensity $I(n_1)$ is defined as

$$I(n_1) = \left| r_1 e^{i(\Phi_1 + \Phi_2)} + r_2 e^{-i(\Phi_1 + \Phi_2)} + r_1 r_2 r_3 e^{i(\Phi_1 - \Phi_2)} + r_2 r_3 e^{-i(\Phi_1 - \Phi_2)} \right|^2 \quad (6)$$

, which accounts for the reflections from each layer and the phase shifts that occur as light traverses the different materials.

Then the contrast C is defined as the relative intensity of reflected light when the DNA is present (n_1) compared to when $n_1 = 1$ (no DNA, i.e., $n_1 = n_0 = 1$). It is calculated as:

$$C = \frac{I_{n_1=1} - I_{n_1}}{I_{n_1=1}} \quad (7)$$

where $I_{n_1=1}$ is the reflected intensity when $n_1 = 1$ (in the absence of DNA), and I_{n_1} is the reflected intensity for the current value of n_1 .

Figure R2 (a) The renormalized reflectance difference, defined as $(R_B - R_S)/R_B$, where R_B and R_S are the reflectance measured at the background and the sample areas, respectively, for the 2D DNA film placed onto a SiO_2 (300 nm)/ Si^{++} substrate, as a function of the wavelength of the perpendicularly incident laser beam. (b) The fitted refractive index $n_1 = n + ik$ of DNA, where the red (blue) line represents the real part n (imaginary part k) of n_1 .

This approach allows us to model the light reflection shown in Figure 2d of our main text, which corresponds to the light contrast C . The fitted n_1 values are presented in Figure R2b, where the red represents the real part of n_1 , denoted as n , and the blue represents the imaginary part of n_1 , denoted as k .

The calculated real part n of the DNA refractive index, as obtained through fitting the above formulas, shows that n has relatively lower values in the wavelength range of $\sim 600\text{--}700$ nm (red visible light region). Correspondingly, the observed optical contrast changes in OM images (shown in Figure 2c of the main text) reveals a reduction in the reflected intensity in the red light range where DNA is present. In contrast, the real part of n is relatively larger in the green light region ($\sim 500\text{--}600$ nm), and the observed optical contrast changes in OM images in this range shows an enhancement in the reflected intensity. The wavelength-dependent behavior of the DNA refractive index may offer insight into the mechanism by which the DNA superlattice interacts with light in the relevant spectral range.

The above analysis are now added in the revised Supplementary Information, in pages 12-13, highlighted in blue.

We thank again Referee#4 for her/his constructive comments that improves the overall quality of our manuscript.

Comment 4. *Figure 2g compares thermal treatment results with and without h-BN. However, it is unclear why the sample without h-BN undergoes degradation upon heating. Is it due to heat-induced deformation of proteins, or is it caused by structural degradation resulting from interactions with specific molecules in the air? A detailed explanation is essential.*

Response:

We appreciate the very careful readings of Referee#4.

We fully agree that a detailed explanation is essential in presenting the Fig. 2g in the main text, which is very helpful for the readers to better comprehend our study.

In the revised manuscript, we carried out extra experiment, as shown in Fig. R3. It is found that when there is absence of moisture (in Fig. R3a pure O₂, or in Fig. R3b pure N₂ environment, for instance), heating at 180 °C for 30 min do not affect too much the morphology of DNA origami structures. However, stark contrast can be seen in the presence of moisture. As can be seen in Fig. R3c, clear degradation of the tessellation morphology is observed after heating the DNA origami in humid air for 30 min. Notice that each image was taken by SEM scan of independent samples, to avoid electron beam exposure prior to heating tests.

The thermal stability of DNA self-assembled structures in aqueous solutions has been extensively studied, with melting temperatures not exceeding 80 degrees Celsius ^[R17]. The primary reason for structural disruption is the intensification of molecular thermal motion, where hydrogen bonds and base stacking that maintain the structural morphology are insufficient to restrain the molecules.

A possible hypothesis is that in a pure gas phase, even if molecules undergo thermal motion, the absence of a diffusion medium makes it difficult for the molecules to diffuse and lose their morphology. However, humid air provides a diffusion medium, which may cause the molecules to lose their structural morphology due to diffusion.

^{R17} Song, J., Arbona, J.M., Zhang, Z., *et al.* *J. Am. Chem. Soc.* **134**, 9844-9847 (2012).

Fig. R3. Thermal stability of DNA origami. SEM images of DNA origami after heating at 180°C for 30 minutes in pure oxygen (a), pure nitrogen (b), and humid air (c). The scale bars in (a-c) are 2 μm .

The above analysis is added in the revised Supplementary Information, in Page 15, highlighted in blue. Description in the main text is also modified, in Page 5, accordingly, also highlighted in blue.

Comment 5. *The occurrence of mini Landau fans due to the superlattice effect is an important result. The authors demonstrate three different superlattice periods, showing that the carrier densities at which mini Landau fans emerge vary accordingly. However, adding the following two aspects would further strengthen the manuscript:*

Response:

1) The key interest here is how the energy band structure of graphene changes with different superlattice periods. While the authors discuss this based on the shift in the positions of the mini Landau fans, incorporating simulation results illustrating the corresponding modifications in the energy band structure would enhance clarity.

We thank the valuable suggestion by Referee#4.

Using previously reported method in similar superlattice system realized by nano-sized etching by focus ion beam^[R18], we construct the following Hamiltonian derived from a continuum model to illustrate the band structure and density of states (DOS) of graphene when the DNA superlattice is present

$$H_{\mu}(\mathbf{k}) = \hbar v_F \mathbf{k} \cdot \boldsymbol{\sigma}_{\mu} + U(\mathbf{r})$$

where $\mu = \pm 1$ is the valley index respectively for \mathbf{K} and $-\mathbf{K}$, with $\boldsymbol{\sigma}_{\mu} = (\mu\sigma_x, \sigma_y)$ denoting Pauli matrices in the sublattice space of graphene. $\mathbf{k} = (k_x, k_y)$ is the 2D wavevector relative to the

^{R18} Barcons Ruiz, D., Herzig Sheinfux, H., Hoffmann, R., *et al.* *Nat. Commun.* **13**, p.6926 (2022).

valley wavevector $\mathbf{K}/-\mathbf{K}$. v_F denotes the Fermi velocity and the value of the parameter is $\hbar v_F = 5.25 \text{ eV} \cdot \text{\AA}$. $U(\mathbf{r})$ refers to the external periodic potential induced from the DNA superlattice, which is expanded in Fourier series truncated to include only the shortest reciprocal lattice vectors of the superlattice

$$U(\mathbf{r}) = V \sum_{m_1 m_2} e^{i(m_1 G_1 + m_2 G_2) \mathbf{r}}$$

where V is the potential amplitude. We solve the eigenvalues of the above Hamiltonian at each \mathbf{k} momentum under a 9×9 plane-wave basis set, which gives 162 electron bands. Here we focus on the low-energy band structure at the \mathbf{K} valley, as the $-\mathbf{K}$ valley structure can be derived through time-reversal symmetry. A dense 300×300 k-point grid is used to ensure the convergence in the DOS calculation. The doping concentration is numerically obtained by the following formula

$$n_{doping} = \frac{1}{N_k \Omega} \int_{E_f}^{E_0} g(E) dE$$

where Ω is the area of superlattice unit cell and N_k is the number of \mathbf{k} points. $g(E)$ is the DOS at E level. E_f is the Fermi energy and E_0 is the energy corresponding to the minimum value of the DOS.

We adopt the potential amplitude $V = 36 \text{ meV}$ when the DNA film is presented in a square superlattice of side length 42 nm . Figure R4a shows the band structure and DOS. Both the electron and hole doping concentration are calculated to be $0.23 \times 10^{12} \text{ cm}^{-2}$. For the DNA film in a hexagonal superlattice of side length 36 nm , with the choice of potential amplitude $V = 20 \text{ meV}$, the doping concentration is calculated to be $0.12 \times 10^{12} \text{ cm}^{-2}$ as indicated in Figure R4b.

These simulation data are in great agreement with the experimental results.

We also note that the selection of potential amplitude is highly flexible as quite a wide range of V values yield consistent doping concentration, which underscores the robustness of the observation in experiment.

Figure R4. Calculated band structure and density of states (DOS) for graphene under the influence of an external potential induced by the DNA film presented in square (a) or hexagonal (b) superlattice. The calculated electron and hole doping concentration are indicated in the energy levels corresponding to the minimum values of the DOS.

The above analysis are updated in the revised manuscript in the Supplementary Note 2, in page 26, highlighted in blue. Description in the main text is also modified, accordingly, also highlighted in blue.

2) Superlattice effects are generally known to become more pronounced under high back-gate voltages, as shown in the ref. 37, 38. To emphasize this aspect, a resistance vs. normalized carrier density plot for different back-gate voltages would significantly improve the argument (you can find in ref. 37, Fig 2 b). Given that the height of the fabricated superlattice (~2 nm) is relatively small, this measurement might be challenging. However, if mini Landau fans are observed, electrical measurements should still be able to capture the effect.

We thank the valuable suggestion by Referee#4, which we totally agree here: *Superlattice effects are generally known to become more pronounced under high back-gate voltages, as shown in the ref. 37, 38.*

Nevertheless, we regret to say that, in our data, although additional Landau fans can be extracted from the fan diagram (Fig. R5a, and Fig. 4 in the main text), the plot of resistance vs. normalized carrier density plot for different back-gate voltages in Fig. R5b-c yields only much weaker signals of the additional peaks (indicated by green arrows), as compared to that reported in such as Ref. 37, Fig 2 b.

Notice that the aligned h-BN moiré (wavelength of ~ 14 nm) does give much higher two side peaks in the same plot, at density slightly higher than $2 \times 10^{12} \text{ cm}^{-2}$. However, for the DNA superlattice (wavelength of $\sim 30\text{-}40$ nm) at the carrier density of about $0.1 \sim 0.2 \times 10^{12} \text{ cm}^{-2}$ (see detailed calculations in Fig. R4), the development of DOS minimum in terms of resistance peak near the central Dirac peak of graphene seems not pronounced at low magnetic fields.

We further attribute this DNA superlattice induced weak signal in resistive peaks at low magnetic fields to the fact that our DNA origami superlattice is currently very thin, it therefore has a weak modulation ability of the gate potential underneath the graphene placed atop, which is indeed insighted by the referee's suggestion as: "*Given that the height of the fabricated superlattice (~ 2 nm) is relatively small, this measurement might be challenging.*" We believe that the fabrication of thicker 2D DNA tessellation in future studies will yield better performance in this regard.

Fig. R5. Magneto-transport behavior of a square DNA/h-BN/graphene/h-BN heterostructure with graphene and h-BN aligned at an angle close to zero degree. (a) Landau fan of the sample, where the black, red, and blue lines represent the main Landau fan of graphene, the mini fan induced by DNA superlattices, and the mini fan induced by the alignment of graphene and BN. (also shown in Fig. 4f in the main text). (b) The evolution of longitudinal resistance (R_{xx}) with magnetic field (B). (c) The evolution of transverse resistance (R_{xy}) with magnetic field (B).

Finally, we would like to sincerely thank the very helpful comments given by Referee #4. The revisions according to her/his suggestions have made our manuscript of much improved quality.

Her/his support in publication in Nature Communications will be greatly appreciated.

Rebuttal Letter

Reviewer #4 (Remarks to the Author):

General Comment. *The authors have addressed all my comments. I believe that the current version of the manuscript is sufficient for publication in Nature Communications.*

Response:

We sincerely appreciate Referee#4 for her/his careful reviewing of our manuscript, and are happy to see the positive feedback “*I believe that the current version of the manuscript is sufficient for publication in Nature Communications.*”

In the coming thread, we will address the remaining questions by the Referee.

Comment 1. *However, I noticed some minor typos in the newly added sections of the Supplementary Information. For example, in Supplementary Information Fig. S10, I found "Fig. R3b," and the sentence "the doping concentration is calculated to be $0.12 \times 10 \text{ cm}^{-3}$ as indicated in Supplementary Figure 19b" seems incorrect, as Supplementary Figure 19b is just an SEM image.*

Before publication, please carefully review the entire Supplementary Information document.

Response:

We have corrected these typos in our Supplementary Information and have thoroughly checked the entire manuscript.

Additionally, some revisions have also been made in response to the editor's suggestions, and all the modifications are highlighted in blue.

Finally, we would like to sincerely appreciate Referee#4 for her/his support of publication in Nature Communications. Her/his very constructive comments have been invaluable in enhancing the quality of our manuscript.